# LANGUAGE MODELS ARE GRAPH LEARNERS

## ABSTRACT

Language Models (LMs) are increasingly challenging the dominance of domain-specific models, including Graph Neural Networks (GNNs) and Graph Transformers (GTs), in graph learning tasks. Following this trend, we propose a novel approach that empowers off-the-shelf LMs to achieve performance comparable to state-of-the-art GNNs on node classification tasks, without requiring any architectural modification. By preserving the LM's original architecture, our approach retains a key benefit of LM instruction tuning: the ability to jointly train on diverse datasets, fostering greater flexibility and efficiency. To achieve this, we introduce two key augmentation strategies: (1) Enriching LMs' input using topological and semantic retrieval methods, which provide richer contextual information, and (2) guiding the LMs' classification process through a lightweight GNN classifier that effectively prunes class candidates. Our experiments on real-world datasets show that backbone Flan-T5 models equipped with these augmentation strategies outperform state-of-the-art text-output node classifiers and are comparable to top-performing vector-output node classifiers. By bridging the gap between specialized task-specific node classifiers and general LMs, this work paves the way for more versatile and widely applicable graph learning models. We will open-source the code upon publication.

## 1 INTRODUCTION

There is a growing trend of utilizing Language Models (LMs) for machine learning tasks across diverse domains. This approach has shown tremendous promise in areas such as computer vision (Desai & Johnson, 2021), audio processing (Mittal et al., 2021), and multimodal learning (Alayrac et al., 2022). In the domain of graph learning, recent efforts have begun to explore the capabilities of LMs in understanding and processing graph structures. Wang et al. (2023) showed that LMs can detect node connectivity and identify cycles within graphs, while Fatemi et al. (2024) explored LMs' ability to evaluate graph scale and identify connected components. Furthermore, InstructGLM (Ye et al., 2023) achieved state-of-the-art performance in text-output node classifiers for Text-Attributed Graphs (TAG) (Zhang et al., 2024a), where each node is associated with textual information.

However, InstructGLM suffers from a fundamental limitation that compromises the generality of the backbone LM. Specifically, InstructGLM expands the LM's vocabulary by creating unique tokens for each node, which incorporates topology-aware node embeddings as token embeddings. While this approach is effective, it comes at the cost of reducing the LM's versatility, making it incompatible with two important use cases: (1) multi-task learning on diverse datasets, a common strategy for training Foundational Models (Wei et al., 2022; Chung et al., 2024), and (2) certain personalized LM fine-tuning services (Li et al., 2024b) that restrict modifications to the backbone model architecture. This limitation raises a crucial question: *How can off-the-shelf, text-to-text instruction-tuned LMs (Raffel et al., 2020) achieve competitive performance in node classification tasks without requiring architectural modifications?*

In stark contrast to (Huang et al., 2023), which suggests that LMs may only interpret graph structures in prompts as contextual paragraphs, our work presents a more optimistic outlook. We aim to overcome this inherent limitation by augmenting the input to LMs while preserving their original architecture. Our proposed model, named AUGGLM (Augmented Graph Language Model), leverages two key augmentation strategies to enhance the LM's ability to process graph data as follows:

- **Relevant Node Retrieval**: In contrast to InstructGLM, which relies on multi-hop ego networks akin to message-passing GNNs for structure-aware contextualization, AUG-GLM draws inspiration from Graph Transformers (GTs) (Min et al., 2022) and Retrieval-Augmented Generation (RAG) (Lewis et al., 2020; Guu et al., 2020). This enables the LM to access long-range structural and semantic information about the target node. We propose two complementary approaches to achieve this: (1) topological retrieval, and (2) prototypical semantic retrieval.

- **Candidate Label Pruning**: To improve LMs' understanding of graph data while maintaining their text-to-text architecture, we convey the guidance from a specialist model, a pretrained lightweight GNN, to the input of LMs via narrowing down the potential labels. This allows LMs to focus on discerning between closely related candidates, ultimately enhancing the performance.

We conduct an extensive evaluation of our approach on four real-world TAGs, showing the effectiveness of our proposed instruction fine-tuning strategy. The results indicate that backbone LMs augmented with AUGGLM significantly outperform InstructGLM, while also matching or surpassing the performance of state-of-the-art vector-output classifiers. These findings represent a crucial step towards bridging the gap between specialized task-specific node classifiers and more general, fine-tuned LMs, highlighting the potential for unified models that can excel across multiple tasks.

## 2 RELATED WORK

**LMs for Graphs**. Recent studies have explored the ability of LMs to understand graph topology by investigating problems such as graph substructure recall (Wang et al., 2024), circle and connectivity detection (Wang et al., 2023; Perozzi et al., 2024), node/edge counting (Perozzi et al., 2024), spatial-temporal problems on dynamic graphs (Zhang et al., 2024b). Notably, Fatemi et al. (2024) found that the presentation format of graph data in text significantly impacts performance across various tasks, highlighting the importance of effective graph-to-text encoding. Building on these findings, several studies have explored classification tasks on TAGs, including node classification (Ye et al., 2023; Zhao et al., 2023b; Li et al., 2024a; Qin et al., 2023), link prediction (Brannon et al., 2023; Tan et al., 2024), transfer learning (Tang et al., 2024), and graph reasoning (Jin et al., 2024); moreover, Chen et al. (2023) present a systematic summary on the performance of off-the-shelf solutions in two categories: LLMs-as-Enhancer and LLMs-as-Predictor. Furthermore, Zhang (2023) proposed Graph-ToolFormer, a framework that enhances LMs with graph reasoning API tools, enabling more complex graph reasoning tasks. GIANT (Chien et al., 2022) and GLEM (Zhao et al., 2023a) utilize the interaction between graph data and LMs to generate better graph representations. TAPE (He et al., 2024) leverages the explanations generated by an LLM to enrich the textual features, which are then used to fine-tune two LMs. The features from these LMs are subsequently passed through an ensemble of GNNs for final prediction.

**Retrieval-Augmented Generation (RAG).** (Lewis et al., 2020; Karpukhin et al., 2020) enhance LMs by granting them access to external knowledge (Hashimoto et al., 2018) during text generation. This technique involves retrieving relevant documents from a large corpus and conditioning the LM on both the input query and the retrieved information. Building on this, REALM Guu et al. (2020) pretrains the retriever and generator end-to-end. Subsequently, RETRO Borgeaud et al. (2022) efficiently scales retrieval-enhanced autoregressive to large datasets. A crucial component of RAG's success is the loss function proposed by Shi et al. (2023) which enables the retriever to be trained with the LM viewed as a black box. HyDE Yu et al. (2023) uses hypothetical document generation to improve retrieval in RAG systems. Furthermore, RAG has extended to multimodal settings Yasunaga et al. (2023). Recently, GraphRAG (Edge et al., 2024) has garnered significant attention which involves constructing a Knowledge Graph (KG) and then generating responses based on the summaries of communities derived from KG. These advancements have significantly improved the factual accuracy and contextual relevance of generated text, solidifying RAG as a promising technique for various applications, including question answering and open-domain dialogue systems.

## 3 PRELIMINARIES

We use the following notation conventions: bold lower-case letters (e.g,. $\mathbf{x}$) denote column vectors, bold upper-case letters (e.g., $\mathbf{X}$) denote matrices, and calligraphic upper-case letters (e.g., $\mathcal{X}$) denote sets. We use $[\cdot]$ and $[\cdot, \cdot]$ to index vectors and matrices, respectively.

We consider the node classification problem on TAGs where each node is associated with textual attributes. A TAG with $n$ nodes is formally represented as $\mathcal{G} = (\mathcal{V}, \mathcal{E}, \mathcal{T})$, where $\mathcal{V} = \{v_i\}_{i=1}^n$ denotes a set of nodes, and $\mathcal{E} = \{e_{ij}\}_{i,j=1}^n$ is a set of edges where $e_{ij} = 1$ indicates that nodes $v_i$ and $v_j$ are connected; otherwise, $e_{ij} = 0$. $\mathcal{T} = \{t_i\}_{i=1}^n$ indicates the set of node textual attributes. The edges can also be represented by an adjacency matrix $\mathbf{A} \in \{0, 1\}^{n \times n}$, where $\mathbf{A}[i, j] = 1$ if and only if $e_{ij} = 1$. The training and test node labels are denoted by $\mathcal{Y} = \mathcal{Y}^{\text{train}} \cup \mathcal{Y}^{test} = \{y_i\}_{i=1}^n$, where each label $y_i$ belongs to one of the $C$ classes, i.e., $y_i \in \{1, \dots, C\}, \forall i$. In the semi-supervised setting considered in this study, we have access to the graph structure and training labels $\mathcal{V}, \mathcal{E}, \mathcal{T}, \mathcal{Y}^{\text{train}}$ during training. The task is to predict the labels of the remaining unlabeled nodes $\mathcal{Y}^{\text{test}}$.

**Personalized PageRank (PPR)** (Page, 1999; Jeh & Widom, 2003) assigns a relevance score to each node in the graph with respect to a given query node. Specifically, given the adjacency matrix $\mathbf{A}$, the PPR scores $\mathbf{r}_i \in \mathbb{R}^n$ for all nodes in the graph relative to the query node $v_i$, are computed iteratively as follows:

$$\mathbf{r}_i \leftarrow (1 - \alpha)\tilde{\mathbf{A}}\mathbf{r}_i + \alpha\mathbf{q}_i \tag{1}$$

where $\alpha \in (0, 1)$ is the teleport probability, $\mathbf{q}_i \in (0, 1)^{n \times 1}$ is a one-hot vector with a single non-zero entry at index $i$, $\tilde{\mathbf{A}} = \mathbf{A}\mathbf{D}^{-1}$ is the normalized adjacency matrix, and $\mathbf{D}$ is the degree matrix. Once the PPR scores are computed, we can identify the top-$K$ most relevant nodes with respect to the query node $v_i$ as follows:

$$\texttt{PPR neigh}(v_i) = \{v_j : \mathbf{r}_i[j] \in \text{top-K}(\mathbf{r}_i)\} \tag{2}$$

**Language Models (LMs).** We employ autoregressive LMs that predict the next token $z_i$ based on the input sequence $t$ and the context of previously generated tokens $z_{1:i-1}$. The probability of generating an $N$-token sequence $z$ given the input sequence $t$ is modeled as:

$$p_{\text{LM}}(z|t) = \prod_{i=1}^N p_{\text{LM}}(z_i|t, z_{1:i-1}) \tag{3}$$

**Retrieval-Augmented Generation (RAG)** (Lewis et al., 2020; Guu et al., 2020) enhances the ability of LMs to answer knowledge-intensive questions. Unlike traditional LMs that directly process the input text $t$, RAG first retrieves a relevant document $d^*$ from an external document corpus $\mathcal{D}$ using a similarity function $s_\phi$:

$$d^* = \arg\max_{d \in \mathcal{D}} s_\phi(d, t) \tag{4}$$

The similarity function $s_\phi$ is typically implemented using a dual-encoder architecture (Bromley et al., 1993) which computes the inner product $\langle \cdot, \cdot \rangle$ between the encoded representations of the document and the input text:

$$s_\phi(d, t) = \langle \text{Encoder}_\phi(d), \text{Encoder}_\phi(t) \rangle \tag{5}$$

Once the relevant document $d^*$ is retrieved, it is fed into the LM along with the initial input $t$ to estimate the output probability $p_{\text{LM}}(z|d^*, t)$. This approach enables LMs to leverage external knowledge and generate more accurate and informative responses.

## 4 METHOD

We explore the application of LMs to node classification tasks in graph learning, leveraging the instruction tuning paradigm to reformulate node classification as a text-to-text task Raffel et al. (2020). Our method employs a carefully designed prompt template and a set of augmentation techniques to transform graph data and ground truth labels into text pairs. This enables LMs to comprehend and be fine-tuned for the task without requiring modifications to their underlying architecture.

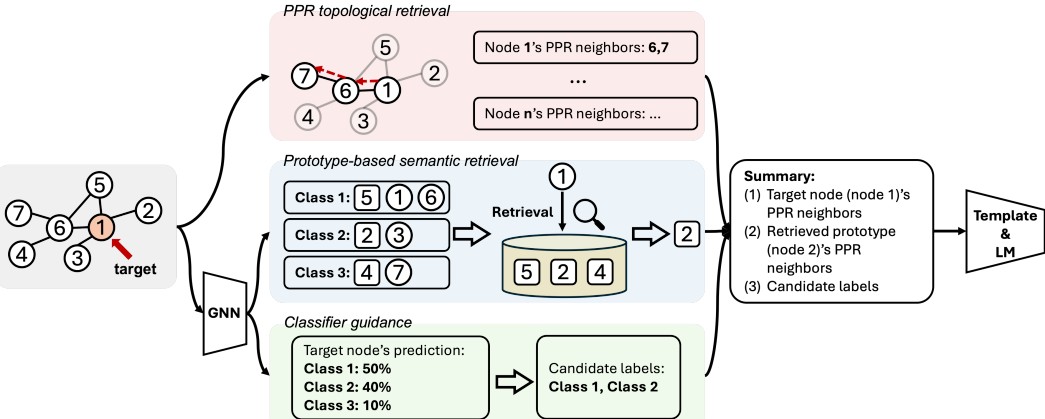

Figure 1: Comparison of pipelines between the existing instruction-tuned LM InstructGLM and our approach, AUGGLM . Unlike InstructGLM, which explicitly encodes graph information into token embeddings as a form of soft prompting, AUGGLM maintains the original text-to-text framework of the off-the-shelf LM, offering greater generality and flexibility.

Figure 2: A detailed pipeline of AUGGLM (ours). In the semantic retrieval module, rectangle nodes denote the prototype of classes.

As shown in Figure 1, our approach differs from InstructGLM (Ye et al., 2023), the current state-of-the-art LM for TAGs, in its underlying design. While both methods utilize prompt templates to transform input graphs into text, InstructGLM relies on explicit encoding of node embeddings into the LM's token embeddings as a form of soft prompting (Lester et al., 2021). In contrast, our approach provides a more general framework, leveraging gradient descent through the LM without modifying its underlying text-to-text architecture. This design choice enables our model to retain the versatility of the original LM while adapting it to graph-based tasks. The following sections provide a detailed description of the specific techniques developed in the paper to achieve this goal.

## 4.1 RETRIEVAL-BASED AGGREGATION

General LMs are not designed to directly process graph-structured data. To overcome this limitation, a common approach is to employ prompt templates that transform graph data and associated tasks into a textual format that LMs can understand. For instance, consider the Cora (Sen et al., 2008) literature citation graph. A typical template (Huang et al., 2023; Ye et al., 2023) for node classification, as shown in Figure 3a consists of three main components: (1) a short description of the classification task, (2) the target node's textual features, such as its title and abstract, and (3) textual features from relevant neighboring nodes within the graph.

The success of the message-passing graph neural networks (GNNs) highlights the importance of the aggregation operation, whose typical examples are the sum and max pooling operations of intermediate node embeddings. A similar spirit is followed for the LM-based classifiers whose key design is at the selection of relevant nodes.

While existing works (Huang et al., 2023; Ye et al., 2023) select 1-hop or multi-hop neighbors as relevant nodes, we posit that this approach is suboptimal for two key reasons. Firstly, not all immediate or extended neighbors provide useful information for classifying the target node, which can

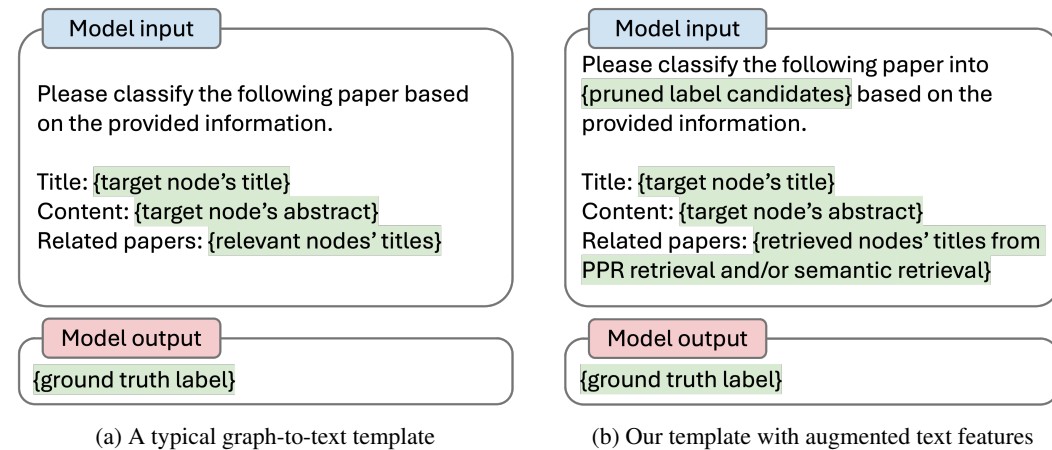

(a) A typical graph-to-text template      (b) Our template with augmented text features

Figure 3: Comparison of a typical graph-to-text template and our template with augmented text features.

introduce noise and degrade model performance. Secondly, incorporating multi-hop neighbors can lead to "neighbor explosion" (Hamilton et al., 2017; Chen et al., 2018; Fey et al., 2021), i.e., an exponentially-growing set of "relevant" nodes, resulting in increased computational costs, slower model training/inference, and even out-of-memory issue. To address these limitations, we propose two novel solutions for selecting relevant nodes: topological retrieval and prototypical semantic retrieval. These methods are designed to efficiently identify the most informative nodes, thereby enhancing the model's ability to capture meaningful graph structures and improve its overall performance.

**Topological Retrieval.** We leverage PPR (Page, 1999; Jeh & Widom, 2003) to perform topological retrieval, which has shown great effectiveness in conjunction with GNNs (Klicpera et al., 2019). The success of PPR suggests that the neighbors it identifies may provide more informative context than generic 1-hop or multi-hop neighbors. Specifically, for a target node $v_i$, we select its top-$K$ neighbors $\text{PPR neigh}(v_i)$ based on their PPR scores, computed using Eqs. (1) and (2). The details of the PPR algorithm are introduced in Section 3. We then concatenate the text features from the PPR neighbors to form the PPR-retrieved text $t^{\text{PPR retri}} = \oplus_{j;v_j \in \text{PPR neigh}(v_i)} t_j$, where $\oplus$ denotes text concatenation.

It is worth noting that the classic PPR algorithm is computationally expensive for large graphs due to the matrix multiplication (Eq. (1)). However, efficient approximate solutions such as Approx-imatePR (Andersen et al., 2006), can be applied to mitigate this issue. Nevertheless, PPR is a topology-based heuristic that does not inherently leverage textual features nor adapt to feedback from downstream node classification tasks. To address these limitations and enhance our framework's semantic awareness, we propose a complementary semantic retrieval strategy, which is discussed in the following section.

**Prototypical Semantic Retrieval.** Our semantic retrieval module draws inspiration from two popular techniques: (1) RAG (Lewis et al., 2020; Guu et al., 2020), which leverage external corpora to answer knowledge-intensive questions, and (2) Graph Transformers (Min et al., 2022) which aggregate messages from distant nodes via inner product-based attention weights. In the context of node classification, we treat the textual features of all nodes except the target node as a surrogate "external corpus." However, unlike typical question-answering tasks, retrieving textual features from a single node is often insufficient for accurate node classification. To address this limitation, we enhance the semantic retrieval process by focusing on prototypes, which capture the essence of each class (Snell et al., 2017).

We employ prototypes (Biehl et al., 2016) as representative examples in the classification problem. To obtain these prototypes, we first pretrain a lightweight GNN $\text{GNN}_\psi$, in a semi-supervised manner. This GNN generates a prediction vector for each node: $\tilde{\mathbf{y}}_i = \text{GNN}_\psi(v_i, \mathcal{G}) \in \mathbb{R}^c, \forall v_i \in \mathcal{V}$. We then

compute the prediction confidence for each node $v_i$ using the maximum logit:

$$\texttt{Conf}(v_i) = \max_j \tilde{\mathbf{y}}_i[j] \in \mathbb{R} \tag{6}$$

The GNN's prediction for node $v_i$ is denoted as $\tilde{y}_i = \arg\max_j \tilde{\mathbf{y}}_i[j] \in \{1, \ldots, C\}$, and the predicted class-$c$ examples are $\tilde{\mathcal{Y}}_c = \{v_i : \tilde{y}_i = c\}$. We select the top-$N$ confident examples as prototypes for each class $c$:

$$\mathcal{P}_c = \left\{ v_i : v_i \in \tilde{\mathcal{Y}}_c \wedge \texttt{Conf}(v_i) \in \text{top-N}\{\texttt{Conf}(v_j) : v_j \in \tilde{\mathcal{Y}}_c\} \right\} \tag{7}$$

This process yields a total of $N \times C$ prototypes: $\mathcal{P} = \bigcup_{c \in \{1, \ldots, C\}} \mathcal{P}_c$. To ensure that we retrieve text features from multiple nodes, we form our corpus $\mathcal{D}$ by concatenating the text features of PPR neighbors for each prototype

$$\mathcal{D} = \left\{ \bigoplus_{j; v_j \in \texttt{PPR neigh}(v_i)} t_j : v_i \in \mathcal{P} \right\} \tag{8}$$

Next, for each target node with its associated text features $t^{\text{target}}$, we compute the semantically retrieved text using Eq. (4): $t^{\text{semantic retri}} = \arg\max_{d \in \mathcal{D}} s_\phi(d, t^{\text{target}})$. In our experiments, we may use topological retrieval, prototypical semantic retrieval, or a hybrid approach that combines both by concatenating their retrieved texts. For simplicity, we denote the retrieved text as $t^{\text{retri}}$.

We defer the discussion of training $\phi$ and $\texttt{GNN}_\psi$ to Section 4.4 and their specific architectures in Section 5.

## 4.2 CLASSIFIER GUIDANCE

Recent studies (Huang et al., 2023; Fatemi et al., 2024; Chen et al., 2024) highlighted the limited understanding of graph topology in current mainstream LMs. While InstructGLM (Ye et al., 2023) addresses this limitation by incorporating node embeddings from a pretrained GNN into the LM's token embeddings, this approach necessitates modifications to the LM's architecture. We propose an alternative method that conveys guidance from a pretrained GNN to inform LMs within the input text space, thereby preserving the LM's original architecture while presenting a pruned set of classification candidates.

We repurpose the pretrained $\texttt{GNN}_\psi$ from the prototypical semantic retrieval module to inform AUG-GLM . For each node $v_i$, we identify and store the top-$I$ predicted labels:

$$\mathcal{L}_i = \{j : \tilde{\mathbf{y}}_i[j] \in \text{top-I}(\tilde{\mathbf{y}}_i)\} \in \{1, \ldots, C\}^I \tag{9}$$

where $I < C$. Given that $\text{IndexToLabel}$ maps are available for our target datasets, which associate numerical labels with their corresponding text representations, we can leverage this mapping to present the pruned label candidates for node $v_i$ as a concatenated text: $t^{\text{candidates}} = \bigoplus_{i \in \mathcal{L}_i} \text{IndexToLabel}(i)$. The integration of this pruned candidate set into the input template is detailed in Section 4.3, where we elaborate on our overall template design.

By adopting this approach, we inject valuable structure-aware inductive bias from the GNN's prediction into the LM's input, thereby enhancing its ability to perform node classification tasks without altering its fundamental architecture. By focusing the LM's attention on a smaller, more relevant set of potential labels, we can improve its classification accuracy.

## 4.3 OVERALL TEMPLATE

Our augmented training samples are presented in Figure 3b, which includes three key elements: (1) the target node's text $t^{\text{target}}$, (2) the retrieved nodes' text $t^{\text{retri}}$, and (3) the pruned label candidates $t^{\text{candidates}}$. We collectively denote these elements as $t^{\text{input}} = (t^{\text{target}}, t^{\text{retri}}, t^{\text{candidates}})$. The backbone LM generates a prediction probability for the label sequence $y^{\text{target}}$ according to the following equation:

$$p_{\text{LM}}(y^{\text{target}} | t^{\text{input}}) = \prod_{i=1}^{|y|} p_{\text{LM}}(y_i^{\text{target}} | t^{\text{input}}, y_{1:i-1}^{\text{target}}) \tag{10}$$

where $|\cdot|$ represents the sequence length; in this equation, $i$ and $1 : i-1$ are token-level indices. We will introduce the detailed selection of the backbone LM in Section 5. Figure 3b presents an exemplar template for the Cora dataset, showcasing the integration of $t^{\text{target}}$, $t^{\text{retri}}$, and $t^{\text{candidates}}$. A full list of templates used on all datasets in our experiments is detailed in Appendix C. Note that we exclude the abstracts of the retrieved nodes to prevent exceeding the maximum input length constraints of most LMs. During evaluation, we utilize only the "model input" portion of this template.

## 4.4 TRAINING

Our framework includes three parameterized modules that require training or fine-tuning: (1) GNNs for generating prototypes and candidate label pruning, as described in Sections 4.1 and 4.2, (2) the encoder $\phi$ from the semantic retriever, defined in Eq. 5, and (3) the backbone LM, utilized in Eq. 10. The GNNs from Sections 4.1 and 4.2 can be shared and their training is independent of the other modules which is supervised by ground truth labels. We provide more details on this process in Appendix A.

For the backbone LM, we adopt a standard training objective: minimizing the average token-wise negative log-likelihood (NLL) between the ground truth target sequence and the model's estimated output probability (Eq. 10). Specifically, for the target node, the NLL loss is computed as:

$$\mathcal{L}_{\text{NLL}}(p_{\text{LM}}(y^{\text{target}}|t^{\text{input}}), y^{\text{target}}) \tag{11}$$

To train the semantic retriever, we employ a distribution-matching loss. Specifically, for a given target node's text feature $t^{\text{target}}$, we first compute the retrieval probability distribution over all prototype text $t \in \mathcal{D}$:

$$p_\phi(t|t^{\text{target}}) = \frac{\exp(s_\phi(t, t^{\text{target}}))}{\sum_{t' \in \mathcal{D}} \exp(s_\phi(t', t^{\text{target}}))} \tag{12}$$

Next, we compute the empirical distribution supervised by the LM as:

$$\tilde{p}_{\text{LM}}(t|t^{\text{target}}, y^{\text{target}}) = \frac{\exp(p_{\text{LM}}(y^{\text{target}}|t^{\text{target}}, t))}{\sum_{t' \in \mathcal{D}} \exp(p_{\text{LM}}(y^{\text{target}}|t^{\text{target}}, t'))} \tag{13}$$

This distribution represents the normalized importance of each prototype text $t \in \mathcal{D}$ based on the LM's likelihood of generating the target ground truth label text $y^{\text{target}}$. We use $\tilde{p}$ to distinguish this distribution from the generation probability defined in Eqs. (3) and (10). For simplicity, we omit the pruned candidate classes $t^{\text{candidates}}$ from the LM input in this equation, although they are indeed included in practice, as shown in Eq. (10).

The distribution matching loss is then computed as the Kullback-Leibler (KL) divergence between the retrieved distribution and the LM-supervised distribution:

$$\text{KL}\left(\text{StopGradient}\left(\tilde{p}_{\text{LM}}\left(\cdot|t^{\text{target}}, y^{\text{target}}\right)\right) \| p_\phi(\cdot|t^{\text{target}})\right) \tag{14}$$

This loss function aims to align the retrieved probability of each prototype text $t \in \mathcal{D}$ with its importance in facilitating the LM's generation of the target label text $y^{\text{target}}$ for the target node. The stop gradient operator ensures that the loss only updates the parameters of the semantic retriever $\phi$, while keeping the LM's parameters $\theta$ frozen. This objective has been used by previous works (Shi et al., 2023; Izacard et al., 2023) without thorough analysis. We provide an in-depth examination of its properties and implications in Appendix B.

Notably, computing Eq. (13) requires $|\mathcal{D}|$ inferences of the backbone LM due to the denominator. However, the LM is fine-tuned only on the NLL loss for the most relevant prototype, $\arg\max_{d \in \mathcal{D}} s_\phi(d, t^{\text{target}})$ via Eq. (11). Consequently, each update step involves $|\mathcal{D}|$ forward passes but only one backward pass. To further reduce the computational overhead associated with $|\mathcal{D}|$ inferences, we can employ a typical sampling strategy: selecting the top-$M$ batch $\mathcal{D}_M = \{t : t \in \text{top-M}_{t' \in \mathcal{D}} s_\phi(t', t^{\text{target}})\}$. By replacing $\mathcal{D}$ with $\mathcal{D}_M$ in Eqs. (12) and (13), we can compute the retrieval probability distribution and the LM-supervised distribution "in-batch", effectively reducing the total number of inferences from $|\mathcal{D}|$ to $M$.

Algorithm 1 outlines a step-by-step process for fine-tuning our entire framework, processing one training node per step. This procedure can be readily extended to mini-batch settings.

---

**Algorithm 1** Training AUGGLM

---

1: **Given**: (1) A graph $\mathcal{G} = (\mathcal{V}, \mathcal{E}, \mathcal{T})$ and training labels $\mathcal{Y}^{\text{train}}$, (2) initialized backbone LM $\theta$, (3) initialized semantic encoder $\phi$, and (4) initialized GNN $\psi$.

2: **Preprocessing**: (1) train the GNN $\psi$ based on $\mathcal{G} = (\mathcal{V}, \mathcal{E}, \mathcal{T})$ and $\mathcal{Y}^{\text{train}}$ till convergence; (2) generate prototypes and their text based on Eqs. (6), (7), and (8); (3) generate the pruned label candidates for every node via Eq. (9).

3: **while** $\theta$ and $\phi$ not converged **do**

4:     Sample $v_i \sim \mathcal{V}$.

5:     Retrieve the relevant nodes' text $t_i^{\text{retri}}$ of node $v_i$ via PPR retrieval (Eq. (2)) and/or semantic retrieval (Eq. (4)).

6:     Plug $t_i$, $t_i^{\text{retri}}$, and $t_i^{\text{candidates}}$ (from preprocessing (3)) into the template (e.g., Figure 3b), compute the NLL loss by Eq. (11), and update $\theta$.

7:     Compute retrieval distribution $p_\phi(\cdot|t_i)$ by Eq. (12).

8:     Call LM inference $|\mathcal{D}|$ times to get $\{p_{\text{LM}}(y_i|t_i, t)\}_{t \in \mathcal{D}}$ and $\tilde{p}_{\text{LM}}(\cdot|t_i, y_i)$.

9:     Compute training loss for retriever by Eq. (14) and update $\phi$.

10: **end while**

---

### 4.5 MODEL COMPLEXITY

Our model consists of three parameterized modules: (1) a GNN $\psi$ for generating prototypes and pruned label candidates, (2) the semantic retriever $\phi$, and (3) the backbone LM $\theta$. Notably, $\psi$ and $\phi$ are lightweight, with a number of parameters that is only $1/30$ to $1/3$ of the number of parameters of LM $\theta$. Compared to the state-of-the-art InstructGLM, our model has an additional module $\phi$, resulting in slightly more parameters which is relatively minor. During training, the GNN $\psi$ can be trained independently, and the PPR scores can be precomputed. The training of $\theta$ relies on the retrieved text from $\phi$, while the training of $\phi$ requires $\tilde{p}_{\text{LM}}(\cdot|t^{\text{target}}, y^{\text{target}})$, which is obtained through forward inference of $\theta$. Importantly, their computational graphs (used for gradient computation) are independent. This is because the LM $\theta$ concatenates the retrieved text from $\phi$ into its input, which is not differentiable with respect to $\phi$. Furthermore, when training $\phi$, the loss in Eq. (14) involves $\tilde{p}_{\text{LM}}(\cdot|t^{\text{target}}, y^{\text{target}})$ wrapped with the StopGradient operator, ensuring that the gradient computation does not update $\theta$. As a result, the cost of back-propagation is similar to updating the LM $\theta$ and the semantic encoder $\phi$ separately.

## 5 EXPERIMENTS

### 5.1 SETUP AND IMPLEMENTATION

Following (He et al., 2024; Ye et al., 2023), we evaluate our approach on four benchmark datasets: Cora (Sen et al., 2008), Pubmed (Sen et al., 2008), ogbn-arxiv (Hu et al., 2020), and a subset of ogbn-products (Hu et al., 2020; He et al., 2024). The statistics of the dataset are summarized in Table 5 (Appendix).

Our implementation employs two pretrained all-MiniLM-L6-v2 models [1] as the dual encoder for the semantic retriever $\phi$ (Eq. (5)) and the text encoder for GNN $\psi$ (Eq. (15)), respectively. We set the teleport probability of the PPR to $\alpha = 0.1$. For the GNN, we employ a 3-layer GraphSAGE (Hamilton et al., 2017) architecture with a hidden dimension of 256 as $\psi$. Our hyperparameter settings include $K = 5$ PPR neighbors, $N = 10$ prototypes, and $M = 8$ samples for LM inference. We choose the number of label candidates $I$, from $\{2, 3\}$. The backbone LM $\theta$ is implemented using Flan-T5-small/base/large (Chung et al., 2022)[2], whose parameters are instruction-fine-tuned using the templates shown in Figure 3b and Section C.

### 5.2 COMPARISON WITH STATE-OF-THE-ART

This section presents the comparison between AUGGLM and the state-of-the-art baselines.

---

[1] https://huggingface.co/sentence-transformers/all-MiniLM-L6-v2

[2] https://huggingface.co/docs/transformers/en/model_doc/flan-t5

Table 1: Performance comparison (accuracy) between AUGGLM and state-of-the-art models. The best-performing vector-output and text-output models are highlighted in **blue** and **red**, respectively.

| | Method | Cora | Pubmed | ogbn-arxiv | ogbn-products |
|---|---|---|---|---|---|
| Vector-output | GCN | 87.78±0.96 | 88.90±0.32 | 73.60±0.18 | 75.64±0.21 |
| | GraphSAGE | 86.51±2.36 | 89.08±0.28 | 73.88±0.33 | 76.04±0.25 |
| | BernNet | 88.52±0.95 | 88.48±0.41 | – | – |
| | FAGCN | 88.85±1.36 | 89.98±0.52 | – | – |
| | GCNII | 88.98±1.33 | 89.80±0.52 | 72.74±0.16 | – |
| | ACM-GCN | 89.75±1.16 | 91.44±0.59 | – | – |
| | GLEM + RevGAT | 88.56±0.60 | 94.71±0.20 | 76.97±0.19 | – |
| | GIANT + RevGAT | 83.53±0.38 | 85.02±0.48 | 75.90±0.19 | 71.89±0.30 |
| | GIANT + GCN | 84.23±0.53 | 84.19±0.50 | 73.29±0.10 | 69.77±0.42 |
| | DeBERTa | 76.06±3.78 | 94.94±0.46 | 73.61±0.04 | 72.97±0.23 |
| | TAPE + RevGAT | **92.90±3.07** | **96.18±0.53** | **77.50±0.12** | **82.34±0.36** |
| Text-output | ChatGPT-3.5 | 67.90 | 93.42 | 73.40 | 74.40 |
| | InstructGLM | 90.77±0.52 | 94.62±0.13 | 75.70±0.12 | – |
| | AUGGLM (T5-small) | 91.14±0.55 | 94.80±0.15 | 75.39±0.21 | 81.73±0.08 |
| | AUGGLM (T5-base) | 91.24±0.46 | 95.03±0.35 | **76.80±0.14** | 81.91±0.11 |
| | AUGGLM (T5-large) | **91.51±0.26** | **95.16±0.18** | 76.00±0.23 | **82.90±0.10** |

We categorize models into two groups: (1) vector-output models which output a vector with dimension equal to the number of classes, and (2) text-output models, which generate text as their output. Specifically, we report results from GCN (Kipf & Welling, 2017), BernNet (He et al., 2021a), FAGCN (Bo et al., 2021), GCNII (Chen et al., 2020), ACM-GCN (Luan et al., 2022), and GLEM (Zhao et al., 2023a)+RevGAT from the leaderboards[345] and their published papers. The results for TAPE+RevGAT, GIANT (Chien et al., 2022)+RevGAT (Li et al., 2021), GIANT+GCN, DeBERTa (He et al., 2021b), and ChatGPT3.5 are reported from (He et al., 2024). The results of InstructGLM are reported from (Ye et al., 2023). Note that all models, except ChatGPT-3.5, are fine-tuned on the training set. We report mean and standard deviation over five runs. For text-output models, we evaluate accuracy by checking whether the model's generated text matches the ground truth label text exactly.

Table 1 presents a comprehensive comparison of our model's performance against existing approaches. The results demonstrate that our proposed method consistently outperforms InstructGLM, achieving new state-of-the-art performance among LMs with text-space outputs for node classification tasks on TAGs. Notably, this superior performance is achieved without modifying the underlying architecture of the LMs, demonstrating the effectiveness of our approach. Furthermore, our models exhibit competitive performance compared to the best vector-output models. Specifically, on Cora, Pubmed, and ogbn-arxiv datasets, our models' performance closely approaches that of the state-of-the-art vector-output models. Furthermore, on the ogbn-products dataset, our approach surpasses the performance of the best vector-output model, TAPE.

## 5.3 ABLATION STUDY

To evaluate the contribution of each key component in AUGGLM , we conducted an ablation study on three crucial modules: (1) topological retrieval, (2) semantic retrieval, and (3) candidate label pruning. We use the Flan-T5-small as the backbone LM for this analysis. The results, presented in Table 2, demonstrate that each module consistently improves performance across all datasets. Notably, our analysis reveals that the relative importance of each component varies across different datasets. For instance, candidate label pruning has a significant impact on performance for the Cora dataset, whereas its effect is less pronounced for the ogbn-products dataset. This variation in com-

---

[3] https://paperswithcode.com/sota/node-classification-on-cora-60-20-20-random
[4] https://paperswithcode.com/sota/node-classification-on-pubmed-60-20-20-random
[5] https://ogb.stanford.edu/docs/leader_nodeprop/

Table 2: Ablation study results. T, S, L, denotes the topological retrieval, semantic retrieval, and candidate label pruning. The ↓ symbol denotes the decrease in accuracy of the ablated version compared to the full model.

| T | S | L | Cora | Pubmed | ogbn-arxiv | ogbn-products |
|---|---|---|------|--------|------------|---------------|
| ✓ | ✓ |   | 85.52 (↓5.62) | 94.40 (↓0.40) | 72.91 (↓2.48) | 79.83 (↓1.90) |
| ✓ |   | ✓ | 87.27 (↓3.87) | 94.32 (↓0.48) | 73.79 (↓1.60) | 81.05 (↓0.68) |
|   | ✓ | ✓ | 90.25 (↓0.89) | 94.26 (↓0.54) | 73.46 (↓1.93) | 79.06 (↓2.67) |
| ✓ | ✓ | ✓ | 91.14 | 94.80 | 75.39 | 81.73 |

Table 3: Performance of the jointly trained model (accuracy).

| Training | Cora | Pubmed | ogbn-arxiv | ogbn-products |
|----------|------|--------|------------|---------------|
| Multi-Task | 91.52 | 94.52 | 74.87 | 82.29 |
| Independent | 91.14 | 94.80 | 75.39 | 81.73 |

ponent importance underscores adaptability of our approach, which can effectively accommodate diverse datasets with different characteristics.

## 5.4 MULTI-TASK TRAINING

One of the key advantage of pure text-to-text instruction tuning is that a single model can be trained on multiple tasks with the same input-output format. To verify this, we conducted an experiment using a Flan-T5-small model, applying our proposed strategies to jointly train it on four diverse datasets: Cora, Pubmed, ogbn-arxiv, and ogbn-products. The results, presented in Table 3 show that the jointly trained model achieve performance comparable to models trained separately on each individual dataset. We observe that on some datasets, such as Cora and ogbn-products, the jointly trained model even outperforms its dataset-specific counterparts.

These findings suggest that our approach can effectively handle multiple graph datasets using a single model, without incurring significant performance losses compared to models trained individually. This capability is crucial for efficient model deployment when dealing with diverse graph data. In contrast, other approaches, such as InstructGLM, require the addition of a large token dictionary to accommodate all nodes in the joint dataset, which hinders their ability to achieve similar generality. Moreover, most vector-output models, including TAPE, are limited by their predefined input-output dimensions, making them inflexible and unable to handle multiple datasets.

## 6 CONCLUSION

We introduce a novel framework called AUGGLM for instruct-tuning Language Models (LMs) to perform node classification tasks on Text-Attributed Graphs (TAGs) using pure text-to-text instructions. Our approach is built upon two key innovations: (1) topological and semantic retrieval of relevant nodes, (2) using a lightweight GNN to guide the classification process of LMs. Extensive experimental results demonstrated the effectiveness of our framework, which consistently outperformed the best existing text-output node classifiers, while achieving performance comparable to state-of-the-art vector-output node classifiers. These findings suggest a promising direction for harnessing the power of Large Language Models (LLMs) in graph learning tasks.

**Limitation and Future Work.** One limitation of this work is the need for manual definition of prompt templates in Table 4. A promising direction for future research is to develop methods for automatically searching for optimal templates in a data-driven manner. Another limitation is the requirement for pretraining a GNN $\psi$ on each dataset, which stems from the inherent challenges of language models in understanding graph data. Addressing this limitation by developing more powerful language models capable of handling graph data is a challenging yet impactful area of future work.

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

## A  ARCHITECTURE AND TRAINING OF THE GRAPH NEURAL NETWORK $\psi$

In our setting, semi-supervised node classification problem, $\mathcal{V}, \mathcal{E}, \mathcal{T}, \mathcal{Y}^{\text{train}}$ are available during training. Since Graph Neural Networks (GNNs) are not inherently capable of processing textual features, we employ a pretrained text encoder to generate $d$-dimensional dense embeddings for each node

$$\text{Encoder}_{\psi_1}(t_i) = \mathbf{h}_i^{(0)} \in \mathbb{R}^d, \forall i \in 1, \dots, n \tag{15}$$

Subsequently, we apply a standard graph neural network. For this study, we adopt Graph-SAGE (Hamilton et al., 2017) whose iterative architecture is

$$\mathbf{h}_i^{(l)} \leftarrow \sigma^{(l)}\Big(\text{MEAN}\Big(\{\mathbf{h}_i^{(l-1)}\} \cup \{\mathbf{h}_j^{(l-1)} : (v_i, v_j) \in \mathcal{E}\}\Big) \cdot \mathbf{W}^{(l)}\Big) \tag{16}$$

where $\sigma^{(l)}$ is the activation function and $\mathbf{W}^{(l)}$ is the learnable parameter of each layer. For an $L$-layed network, in the last layer, $\sigma^{(L)}$ is selected as Softmax and $\mathbf{W}^{(L)} \in \mathbb{R}^{d \times c}$ resulting in $\mathbf{h}_i^{(L)} \in \mathbb{R}^c$ as the prediction vector. The typical loss used for training the GNN is negative log-likelihood $\mathcal{L}_{\text{NLL}}(\mathbf{h}_i^{(L)}, y_i)$ for all the nodes in the training set $\mathcal{Y}^{\text{train}}$. The complete set of trainable parameters is denoted as $\psi = \{\psi_1\} \cup \{\mathbf{W}^{(l)}\}_{l=1}^L$.

## B  INTERPRETATION OF THE DISTRIBUTION MATCHING LOSS

We recap the objective function. For notation brevity, we use $t_i$ to denote the input target node $t^{\text{target}}$:

$$\text{KL}(\tilde{p}_{\text{LM}}(\cdot|t_i, y_i)\|p_\phi(\cdot|t_i)) \tag{17}$$

where the stop gradient operator is removed if we only compute gradient with respect to $\phi$ and

$$p_\phi(t_j|t_i) = \frac{\exp(s_\phi(t_i, t_j))}{\sum_{t_k \in \mathcal{D}} \exp(s_\phi(t_i, t_k))} \tag{18}$$

and

$$\tilde{p}_{\text{LM}}(t_j|t_i, y_i) = \frac{\exp(p_{\text{LM}}(y_i|t_i, t_j))}{\sum_{k \in \mathcal{N}_i} \exp(p_{\text{LM}}(y_i|t_i, t_k))} \tag{19}$$

For notation brevity, we replace $\sum_{t_k \in \mathcal{D}}$ with $\sum_z$ if there is no ambiguity. Then

$$\min_\phi \text{KL}\big(\tilde{p}_{\text{LM}}(\cdot|t_i, y_i)\|p_\phi(\cdot|t_i)\big) \Leftrightarrow \min_\phi -\sum_z \tilde{p}_{\text{LM}}(z|t_i, y_i) \log[p_\phi(z|t_i)] \tag{20}$$

$$= -\sum_z \tilde{p}_{\text{LM}}(z|t_i, y_i) \log\left[\frac{e^{s_\phi(z, t_i)}}{\sum_{z'} e^{s_\phi(z', t_i)}}\right] \tag{21}$$

$$= \sum_z \tilde{p}_{\text{LM}}(z|t_i, y_i) \log\left[\sum_{z'} e^{s_\phi(z', t_i)}\right] - \sum_z \tilde{p}_{\text{LM}}(z|t_i, y_i) s_\phi(z, t_i) \tag{22}$$

$$= \log\left[\sum_z e^{s_\phi(z, t_i)}\right] - \sum_z \tilde{p}_{\text{LM}}(z|t_i, y_i) s_\phi(z, t_i) \tag{23}$$

Hence,

$$\nabla \text{KL} = \frac{\sum_z e^{s_\phi(z, t_i)} \nabla s_\phi(z, t_i)}{\sum_{z'} e^{s_\phi(z', t_i)}} - \sum_z \tilde{p}_{\text{LM}}(z|t_i, y_i) \nabla s_\phi(z, t_i) \tag{24}$$

$$= \sum_z \left[p_\phi(z|t_i) - \tilde{p}_{\text{LM}}(z|t_i, y_i)\right] \nabla s_\phi(z, t_i) \tag{25}$$

$$= \sum_z \left[1 - \frac{\tilde{p}_{\text{LM}}(z|t_i, y_i)}{p_\phi(z|t_i)}\right] p_\phi(z|t_i) \nabla s_\phi(z, t_i) \tag{26}$$

Table 4: Templates used for all datasets.

| Template name | Text |
|---|---|
| Citation
(for Cora, Pubmed, ogbn-arxiv) | Please classify the following paper into {pruned label candidates} based on the provided information\nTitle: {target node's title}\nContent: {target node's abstract}\nRelated papers: {retrieved nodes' titles} |
| Citation Title Last
(for Cora, Pubmed, ogbn-arxiv) | Please classify the following paper into {pruned label candidates} based on the provided information\nContent: {target node's abstract}\nRelated papers: {retrieved nodes' titles}\nTitle: {target node's title} |
| Amazon
(for ogbn-products) | Please classify the following Amazon product into {pruned label candidates} based on the provided information\nProduct name: {target node's title}\nDescription: {target node's description}\nRelated products: {retrieved nodes' titles} |
| Amazon Title Last
(for ogbn-products) | Please classify the following Amazon product into {pruned label candidates} based on the provided information\nDescription: {target node's description}\nRelated products: {retrieved nodes' titles}\nProduct name: {target node's title} |

Table 5: Dataset statistics.

| Name | # nodes | # edges | # classes | Split | Metric |
|---|---|---|---|---|---|
| Cora | 2,708 | 10,556 | 7 | Random 60/20/20% | Accuracy |
| Pubmed | 19,717 | 88,648 | 3 | Random 60/20/20% | Accuracy |
| ogbn-arxiv | 169,343 | 1,166,243 | 40 | Given split | Accuracy |
| ogbn-products | 54,025 | 198,663 | 47 | Given split | Accuracy |

After changing the notation back from $\sum_z$ to $\sum_{t_k \in \mathcal{D}}$, we have

$$\nabla \text{KL} = \sum_{t_k \in \mathcal{D}} \left[ 1 - \frac{\tilde{p}_{\text{LM}}(t_j|t_i, y_i)}{p_\phi(t_j|t_i)} \right] p_\phi(t_j|t_i) \nabla s_\phi(t_j, t_i) \tag{27}$$

whose rationale is that if the LM's feedback greatly prefers the neighbor $v_j$ (and its associated text $t_j$), larger than its probability to be retrieved by the retriever (i.e., $\frac{\tilde{p}_{\text{LM}}(t_j|t_i, y_i)}{p_\phi(t_j|t_i)} > 1$), then the similarity score between $t_i$ and $t_j$ will increase, i.e., improve the probability of $t_j$ to be retrieved.

## C  TEMPLATES

Table 4 presents templates used in this paper. We design the "Citation" template for the Cora, Pubmed, and ogbn-arxiv datasets and the "Amazon" template for the ogbn-products dataset.

Drawing inspiration from the findings of He et al. (2024), who demonstrated the efficacy of positioning the title after the main content for certain datasets, we have also introduced two additional template variations: "Citation Title Last" and "Amazon Title Last."

## D  DATASET STATISTICS

We present the detailed statistics of datasets used in this paper in Table 5.

# E ADDITIONAL EXPERIMENTS

## E.1 ADDITIONAL EFFICIENCY STUDY

**FLOPs** Here we compare the floating point operations (FLOPs) of our proposed AUGGLM and our main baseline, InstructGLM Ye et al. (2023), a text-output node classifier.

The computation of InstructGLM includes (1) computing node encodings, which is precomputed, and (2) training and inference of the downstream LM. In contrast, the computation of our AugGLM includes (1) computing PPR neighbors for every node, which is also precomputed, (2) training and inference of the semantic retriever, and (3) training and inference of the downstream LM. Hence, the extra on-the-fly computation overhead comes from the semantic retriever, all-MiniLM-L6-v2, in our experiments. We report the FLOPs of the retriever and different LM backbones in Table 6.

Table 6: FLOPs comparison between different modules.

|                  | FLOPs  |
| ---------------- | ------ |
| Retriever        | 2.3G   |
| FLAN-T5 (small)  | 71.7G  |
| FLAN-T5 (base)   | 257.2G |
| FLAN-T5 (large)  | 845.4G |

The results show that the retriever only adds a tiny amount of FLOPs compared to the backbone LMs. In other words, the FLOPs of our framework are very close to those of InstructGLM if we adopt the same downstream LM. More concretely, if both our model and InstructGLM select T5-large as the backbone, the FLOPs of InstructGLM would be 845.4G, and our framework would be 847.7G.

**Memory usage.** Memory usage is linear concerning batch size. We report the memory usage with different backbone LMs in Table 7, where we set the batch size to 1.

Table 7: Memory usage with different LMs.

|                  | GPU memory |
| ---------------- | ---------- |
| AugGLM (small)   | 3098M      |
| AugGLM (base)    | 6572M      |
| AugGLM (large)   | 20308M     |

It is reasonable that more powerful backbone LMs require more GPU memory.

**Convergence analysis.** We train AUGGLM with different backbones: FLAN-T5-small/base/large on the Cora dataset and plot their loss curve regarding updating steps in Figure 4. It shows that our proposed AUGGLM is easy to train and converges smoothly when equipped with various backbones LMs of different scales.

**Running time.** We recorded the running time (both forward and backpropagation) of the semantic retriever and the backbone LMs in Table 8. The dataset we tested was Cora, and the batch size was 1. Note that this wall clock running time is related to the batch size, dataset, and specific hardware. In this experiment, the running time is tested on an NVIDIA A100-SXM4-80GB.

Overall, we can conclude that the semantic retriever only adds very limited on-the-fly computation overhead compared to the downstream LM, showing the efficiency of our proposed framework.

## E.2 ADDITIONAL PARAMETER STUDY

In this section, we study the model's performance with various hyperparameters.

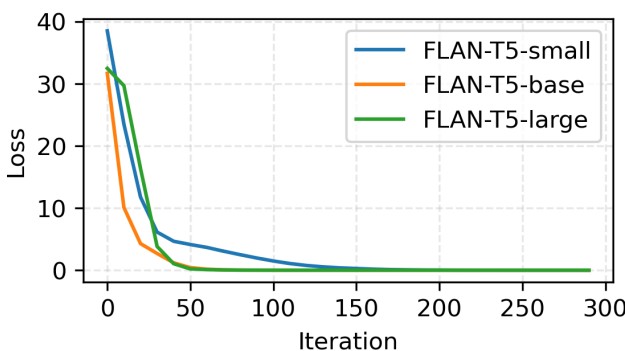

Figure 4: Convergence curve of AUGGLM .

Table 8: Wall-clock running time comparison between different modules.

|                  | Forward (ms) | Backprop (ms) |
|------------------|--------------|---------------|
| Retriever        | 14.7         | 6.1           |
| FLAN-T5 (small)  | 90.0         | 32.0          |
| FLAN-T5 (base)   | 104.4        | 66.6          |
| FLAN-T5 (large)  | 277.2        | 197.0         |

**Selection of the backbone GNN.** Specifically, we study the performance of AUGGLM equipped with different GNNs. we compared the performance of AugGLM equipped with GraphSAGE (used in the reported results) with the counterpart equipped with GCN Kipf & Welling (2017). The comparison is in Table 9.

Table 9: Performance comparison of AUGGLM equipped with different GNNs.

|           | Cora  | Pubmed | ogbn-arxiv | ogbn-products |
|-----------|-------|--------|------------|---------------|
| GraphSAGE | 91.14 | 94.80  | 75.39      | 81.73         |
| GCN       | 90.98 | 94.85  | 75.21      | 81.82         |

We observed that the performance is nearly identical between GCN and GraphSAGE. This can be attributed to two factors: (1) the classification performances of GCN and GraphSAGE are similar, and (2) the GNN is used to generate prototypes and prune candidate labels, which does not require a highly powerful GNN for accurate classification.

**Number of PPR retrieved nodes.** Next, we preliminarily examined the relationship between the model performance and the number of nodes retrieved. In this auxiliary experiment, we fixed the number of nodes retrieved by semantic retrieval at 5 and varied the number of nodes retrieved by PPR retrieval. The results are reported in Table 10

Table 10: Performance comparison of AUGGLM with different PPR retrieved neighbors.

| PPR neighbors | 1     | 3     | 5     | 7     | 9     | 10    | 15    | 20    | 25    |
|---------------|-------|-------|-------|-------|-------|-------|-------|-------|-------|
| ogbn-arxiv    | 75.18 | 75.76 | 75.39 | 75.19 | 76.05 | 76.45 | 75.99 | 74.81 | 74.48 |

Interestingly, we found that the model's performance remains relatively stable when the number of PPR nodes is less than 15. However, the performance degrades when too many nodes are retrieved (more than 15). A possible explanation is that when the number of PPR nodes becomes too large, every target node's retrieved nodes become similar (e.g., some hub nodes are retrieved by most nodes), reducing the discriminativeness of each target node. This phenomenon is reminiscent of the

"oversmoothing" problem Li et al. (2018) in GNNs, where a GNN with too many layers produces indistinguishable latent representations for all nodes.

**Other topological retrieval options.** In this auxiliary experiment, we use the link predictor to retrieve relevant neighbors. Specifically, we trained a graph autoencoder (GA) Kipf & Welling (2016), a basic graph neural network-based link predictor, on the given graph. Then, we retrieved the top-5 most confident neighbors from the reconstructed graph to replace those obtained through PPR retrieval* The results are presented in Table 11, where Flan-T5-small is used as the backbone. For better reference, we also provide a version where PPR retrieval is replaced with retrieving from 1-hop neighbors.

Table 11: Performance comparison of AUGGLM with different topological retrieval techniques.

|                | Cora  | Pubmed | ogbn-arxiv | ogbn-products |
|----------------|-------|--------|------------|---------------|
| 1-hop neighbors | 90.59 | 94.33  | 73.97      | 79.53         |
| GA             | 90.83 | 94.42  | 74.01      | 79.85         |
| PPR neighbors  | 91.14 | 94.80  | 75.39      | 81.73         |

We observe that both 1-hop neighbor retrieval and GA perform worse than their PPR counterparts. A possible reason is that both 1-hop neighbor retrieval and GA are local retrieval methods, whereas PPR can effectively capture the global structure. Additionally, we note that GA is trained using a reconstruction loss, which means it tends to assign high confidence to existing edges. In other words, the neighbors retrieved by GA would be similar to those obtained through 1-hop neighbor retrieval, except for some low-degree nodes.

# F SELECTED HYPERPARAMETERS

We report the hyperparameter used for every dataset in Table 12. More detailed hyperparameters will be released with the code upon publication.

Table 12: Hyperparameters selected of AUGGLM .

|                        | Cora     | Pubmed   | ogbn-arxiv         | ogbn-products |
|------------------------|----------|----------|--------------------|---------------|
| # PPR neighbors        | 5        | 2        | 5                  | 5             |
| # Semantic neighbors   | 5        | 2        | 5                  | 5             |
| Template               | Citation | Citation | Citation title last | Amazon        |
| # Candidate labels     | 3        | 2        | 3                  | 3             |
| LM learning rate       | 1e-4     | 1e-4     | 1e-4               | 1e-4          |
| Retriever learning rate | 1e-5     | 1e-5     | 1e-5               | 1e-5          |
| Weight decay           | 0        | 0        | 0                  | 0             |

