# OpenReview forum: "Language Models are Graph Learners"
_ICLR.cc/2025/Conference — Submitted to ICLR 2025_

### Official Review · Reviewer_MmYP · 2024-10-23

**Soundness:** 3
**Presentation:** 2
**Contribution:** 2
**Rating:** 5
**Confidence:** 4

**Summary:**

This paper proposes a new methodology called 'AUGGLM (Augmented Graph Language Model)'. This approach enhances node classification performance without modifying existing language models (e.g., Flan-T5). The primary idea is to enrich the input to the language model using topological and semantic search methods, enabling better graph data processing. Additionally, a lightweight Graph Neural Network (GNN) is used to prune candidate labels, improving classification performance. Through these augmentation techniques, the model demonstrates performance that either surpasses or matches existing text-based node classifiers, with the goal of developing a model applicable to various tasks.

**Strengths:**

1. AUGGLM can process graph data without modifying the existing language model architecture, making it easily applicable to various tasks.
2. AUGGLM performs on par with traditional Graph Neural Networks (GNNs), and in certain datasets, it achieves even better results.

**Weaknesses:**

1. Although various engineering techniques were applied to contribute to performance improvement, they are merely a combination of existing, publicly available techniques, with no original or novel methods included.
2. It is unclear how the method presented in Section 4.2 differs from widely known ensemble techniques. It seems to be a simple ensemble of GNN and language models (LM), and the unique contributions of the authors are not clearly explained.
3. There is a lack of clear motivation for each module—Topological Retrieval, Prototypical Semantic Retrieval, and Classifier Guidance—and an insufficient explanation of the theoretical background for using these three modules together. It gives the impression of a system that compiles widely known techniques.

**Questions:**

1. The overall process is quite complex. Have diagrams been prepared for each module—Topological Retrieval, Prototypical Semantic Retrieval, and Classifier Guidance?
2. While the motivation for each module is indirectly addressed, the clear rationale for using all three modules together has not been fully explained. Could you provide additional explanation regarding the theoretical background for proposing these three modules together?
3. Classifier Guidance appears to be a simple ensemble technique rather than an attempt to effectively utilize the language model (LM). Furthermore, without the ensemble technique, it would be difficult to claim superior performance over InstructGLM. Is it merely a simple ensemble method? If not, could you explain in detail the novel aspects uniquely proposed by the authors?

---

> ### Author Response · Authors · 2024-11-21
> **Response to Reviewer MmYP (1/2)**
>
> Dear reviewer MmYP, We appreciate your tremendous time and effort in reviewing our paper. We thank you for recognizing that our method is applicable to various tasks and that its performance is good. For your questions, our point-to-point responses are as follows.
>
> ---
>
> > **Q1.** Although various engineering techniques were applied to contribute to performance improvement, they are merely a combination of existing, publicly available techniques, with no original or novel methods included.
>
> **A1.** We agree with the reviewer that certain techniques, such as Personalized PageRank and semantic retrieval, are well-established. However, we would like to highlight the following unique novelties and contributions of this paper:
> 1. Traditional (trainable) retrieval-augmented generation (RAG) methods can only retrieve **one related document** for each sample during LM training and inference. This is insufficient for node classification tasks, which require **textual features from multiple connected nodes**. To address this limitation, we propose prototype-based semantic retrieval, which can retrieve textual features from multiple nodes while maintaining a **trainable** retriever. To our best knowledge, **this strategy is novel and has not been studied before, particularly in the context of node classification tasks**. The most similar technique to our prototype-based retrieval is GraphRAG [1], which relies heavily on the power of LLM to construct a knowledge graph and retrieve a subset of the external corpus. Our proposed solution is distinct from this approach.
>
> 2. While the KL loss used for training the retriever based on feedback from the LM has been proposed in existing efforts, it **has not been thoroughly analyzed**. We analyzed the training process using this loss (Lines 361-363).
>
> 3. The classifier guidance technique presents a unique novelty: it **achieves the cascade of two vastly different models (a GNN and an LM) in the text space**. In traditional deep learning solutions, different modules typically cascade in the latent space. However, in the era of LMs, **if we aim to avoid modifying the pre-trained LM architecture**, cascading the GNN and LM in the text space is a more useful technique. To the best of our knowledge, the closest technique to our classifier guidance is Mixture-of-agents [7], which cascades multiple LMs in the text space. Our proposed solution is clearly distinct from this approach.
>
> ---
>
> > **Q2.** It is unclear how the method presented in Section 4.2 differs from widely known ensemble techniques. It seems to be a simple ensemble of GNN and language models (LM), and the unique contributions of the authors are not clearly explained.
>
> **A2.** We appreciate the reviewer's insightful question and are pleased to provide clarification.
>
> 1. Our GNN classifier guidance is **not** an ensemble method [2]. A typical ensemble method **combines the outputs of multiple models** to produce the final output. In contrast, our framework's final output comes **solely from a single LM**.
>
> 2. Our classifier guidance technique works **sequentially**: first, the GNN predicts and prunes the labels to obtain a set of candidate labels; then, these candidate labels are incorporated into the LM's input for accurate final classification.
>
> 3. As we mentioned in our response to Q1, this technique possesses a **unique novelty** in that it **achieves the cascade of two vastly different models (a GNN and an LM) in the text space**. This approach is particularly useful in the era of LMs, as it allows us to avoid modifying the pre-trained LM architecture.

---

> ### Author Response · Authors · 2024-11-21
> **Response to Reviewer MmYP (2/2)**
>
> > **Q5.** While the motivation for each module is indirectly addressed, the clear rationale for using all three modules together has not been fully explained. Could you provide an additional explanation regarding the theoretical background for proposing these three modules together?
>
> **A5.** Please check our response to Q3.
>
> ---
>
> > **Q6.** Classifier Guidance appears to be a simple ensemble technique rather than an attempt to effectively utilize the language model (LM). Furthermore, without the ensemble technique, it would be difficult to claim superior performance over InstructGLM. Is it merely a simple ensemble method? If not, could you explain in detail the novel aspects uniquely proposed by the authors?
>
> **A6.** Please check our response to Q2.
>
>
> ```
> Reference:
>
> [1] Edge, Darren, et al. "From local to global: A graph rag approach to query-focused summarization." arXiv 2024.
>
> [2] https://www.ibm.com/topics/ensemble-learning
>
> [3] Park, Sungchan, et al. "A survey on personalized PageRank computation algorithms." IEEE Access 7 (2019): 163049-163062.
>
> [4] Grover, Aditya, and Jure Leskovec. "node2vec: Scalable feature learning for networks." SIGKDD 2016.
>
> [5] Andersen, Reid, Fan Chung, and Kevin Lang. "Local graph partitioning using pagerank vectors." FOCS 2006.
>
> [6] Min, Erxue, et al. "Transformer for graphs: An overview from architecture perspective." arXiv 2022.
>
> [7] Wang, Junlin, et al. "Mixture-of-Agents Enhances Large Language Model Capabilities." arXiv 2024.
> ```

---

> > ### Author Response · Authors · 2024-11-24
> > **Supplementary response**
> >
> > We apologize that a part of the response is mistakenly missing. We supplement them here.
> >
> > > **Q3.** There is a lack of clear motivation for each module—Topological Retrieval, Prototypical Semantic Retrieval, and Classifier Guidance—and an insufficient explanation of the theoretical background for using these three modules together. It gives the impression of a system that compiles widely known techniques.
> >
> > **A3.** We are pleased to provide further clarification on our approach's motivation and theoretical background.
> >
> > 1. Personalized PageRank (PPR) topological retrieval has been extensively studied and proven to be an effective solution for **retrieving relevant nodes** [3]. Its ability to explore the graph's global structure offers empirical advantages over methods considering only $1$-hop neighbors [4]. Also, approximate solutions for PPR are well-developed [5], allowing us to precompute it efficiently **without incurring additional computational overhead** during downstream LM training. Therefore, we have incorporated PPR into our framework.
> >
> > 2. One limitation of topological retrieval is that it overlooks the textual features of each node. To address this, we propose the semantic retrieval module as a **supplementary** component. Our semantic retrieval module draws inspiration from **graph Transformer/self-attention** modules [6]: a self-attention layer computes an attention matrix to aggregate representations from other nodes, and **the normalized similarity score (Eq. 12) used by our semantic retrieval module can be viewed as an attention weight**. Based on this similarity score, the semantic retrieval module **adaptively selects relevant nodes** to enhance the context of the target node.
> >
> > 3.  The classifier guidance was introduced to overcome **a practical challenge** we encountered in our experiments. When inputting all candidate labels into the LM, such as "please classify the above research paper into class 1, class 2, ..., class n," **the list of candidate labels can become excessively long** for certain datasets (e.g., 47 classes for the products dataset). This lengthy input slows down the LM's training and increases its search space, leading to suboptimal performance. To mitigate this issue, we pretrain a GNN to prune the label space, resulting in a **significantly shorter input** for the LM and **improved accuracy**.
> >
> > ---
> >
> > > **Q4.** The overall process is quite complex. Have diagrams been prepared for each module—Topological Retrieval, Prototypical Semantic Retrieval, and Classifier Guidance?
> >
> > **A4.** Thank you for suggesting this improvement. In the **revised Figure 2**, we have carefully illustrated a detailed pipeline of all the modules used in our framework, which we expect will enhance readers' understanding of our proposed framework.
> >
> > ---
> >
> > As always, we are happy to address any concerns you might have regarding our paper.
> >
> > Sincerely,
> >
> > Authors

---

> ### Comment · Reviewer_MmYP · 2024-11-22
>
> Thank you for your response. It address my concerns.

---

> > ### Author Response · Authors · 2024-11-22
> >
> > We are glad to hear that your concerns have been addressed. Feel free to let us know if you have any further questions; we will happily answer them.
> >
> > Authors

---

### Official Review · Reviewer_dzLA · 2024-10-26

**Soundness:** 2
**Presentation:** 3
**Contribution:** 2
**Rating:** 5
**Confidence:** 4

**Summary:**

This paper introduces an approach that enables off-the-shelf Language Models (LMs) to perform node classification tasks on Text-Attributed Graphs (TAGs) with a performance comparable to state-of-the-art Graph Neural Networks (GNNs).  Authors propose two strategies for enriching LM inputs and the proposed method empowers LMs to handle graph data without altering their original architecture, thus maintaining the LMs' versatility for multi-task learning and other applications. Also, authors propose to use a GNN-based classifier to guide the training of LMs, which enhances the efficiency of model prediction.

**Strengths:**

* Motivation for making the model capable of processing graph data without changing the model architecture is interesting and proposes a reasonable approach
* Paper is well structured, with a clear introduction,  methodology, and experiments, making it easy for the reader to follow the author's thoughts.
* An experimental evaluation of real-world data sets proves the effectiveness of the proposed method. The use of multiple datasets increases the reliability of the results.

**Weaknesses:**

* Incorporation of structural information through the addition of related nodes appears to be a well-established concept, I doubt the source of the performance improvement is, see the question section.
* The scalability of the model is limited, increasing the number of input nodes will inevitably increase the length of input text, which may be intractable in some situations, and the paper lacks discussions for this.
* Model performance is limited by the GNN model. For example, if the GNN model cannot give accurate candidate labels, then the LM model cannot give correct predictions.
* Experiments are thin and lack detailed additional experiments and analysis, such as hyperparameters.

**Questions:**

* As mentioned in weakness, the proposed method is similar to few-shot methods, can you provide comparison between your method and some simple baselines, such as retrieval neighbor nodes or high text-similarity nodes to enhance the input?
* Whether it can provide the performance of the model related to key hyperparameters, such as the number of nodes retrieved or different GNN models, is critical to understanding the key components of the method and the scalability.
* Most LM-based methods are inductive. Is the proposed method inductive? Have you tried zero-shot experiments? For example, training on one dataset. and test on other datasets.
* Why wasn't the experiment conducted on decoder-only LLMs such as 7b model, which would have enhanced the integrity and significance of the paper.  Although it doesn't change my rating on the paper.

---

> ### Author Response · Authors · 2024-11-21
> **Response to Reviewer dzLA (1/3)**
>
> We appreciate the tremendous time and effort you took to review our paper. Thank you for recognizing that our method is reasonable, that our paper is well-organized, and that our experiments are reliable. Below, we provide point-by-point responses to your questions.
>
>
> ---
>
> > **Q1.** The scalability of the model is limited, increasing the number of input nodes will inevitably increase the length of input text, which may be intractable in some situations, and the paper lacks discussions for this.
>
> **A1.** We agree with the reviewer that retrieving relevant nodes may lead to increased input length, slower training times, or even reaching the limit of input windows. To address this concern, we would like to offer the following perspectives:
>
> 1. One of the primary motivations behind our proposed topological and semantic retrieval (Lines 215-224) is to **wisely** select a limited number of relevant nodes. For instance, in our experiments, we only retrieve $10$ relevant nodes ($5$ from topological retrieval and $5$ from semantic retrieval).
>
> 2. To prevent excessive input length growth, **only the title** from the retrieved nodes are included (Lines 306-308).
>
> 3. In fact, our framework is **more efficient than the SOTA LM-based classifier [1]**.  [1] mentions in their limitation section that *"the instruction prompts we construct may not encompass all high-order neighbors within a single natural language sentence due to the limitations of sentence length"*. This implies that our topological retrieval and semantic retrieval can more effectively identify relevant nodes **without overwhelming the input window of the LM with a large number of neighbor nodes**.
>
> We appreciate the reviewer's insightful comments, which led to this meaningful discussion. We will incorporate these points into the revised paper.
>
> ---
>
> > **Q2.** Model performance is limited by the GNN model. For example, if the GNN model cannot give accurate candidate labels, then the LM model cannot give correct predictions.
>
> **A2.** We agree with the reviewer that our framework's effectiveness will degrade if the GNN's prediction is poor. However, **the accuracy requirement of the backbone GNN is not overly strict** because its primary goal is to **provide a set of reliable candidates, rather than an accurate class label**. For example, on the ogbn-arxiv dataset, a simple GCN achieves an **accuracy of ~$72-73%$%**, but the probability that the groundtruth is among **the top-3 candidates** predicted by that GCN can be as high as **$94$%**. In other words, even if the GNN is not particularly powerful, it can still effectively narrow down the candidate labels for the downstream LM.
>
> Additionally, we compare performance using different GNN backbones in our response to your Q4, which further verifies that our framework's requirement for the choice of GNN is not overly strict, even if they are not highly powerful.
>
>
> ---
>
>
> > **Q3.** As mentioned in weakness, the proposed method is similar to few-shot methods, can you provide comparison between your method and some simple baselines, such as retrieval neighbor nodes or high text-similarity nodes to enhance the input?
>
> **A3.** Yes. We have conducted additional experiments to evaluate our framework further. Specifically, we followed the setting of our ablation study (Section 5.3) and replaced certain modules with simpler baselines.
>
> 1. **Firstly, we replaced PPR retrieval with a simple 1-hop neighbor retrieval.** In this approach, $K$ neighbors are uniformly sampled for nodes with more than $K$ neighbors. We set $K=5$, consistent with the number of PPR neighbors used in our proposed method. Current LM-based node classifiers, such as [1], also use this strategy.
>
> 2. **Secondly, we replaced our prototype-based semantic retrieval with a simple retriever that selects the K-most textually similar nodes.** Again, we set $K=5$, matching the number of PPR neighbors used for each prototype in our original method. Notably, this simple semantic retriever is **not amenable to fine-tuning based on the feedback from downstream LM model**, due to the vast number of possible combinations; allowing the LM to make inferences for every possible combination would be computationally intractable.
>
> |  | Cora | Pubmed | ogbn-arxiv | ogbn-products |
> |---|---|---|---|---|
> | 1-hop neighbor retrieval | 90.59 | 94.33 | 73.97 | 79.53 |
> | Simple semantic retrieval | 90.68 | 94.37 | 74.46 | 81.21 |
> | AugGLM | 91.14 | 94.80 | 75.39 | 81.73 |
>
> Our results show that: (1) the simple $1$-hop neighbor retrieval performs worse than PPR retrieval because it is limited to **local** exploration and **lacks global understanding** of the graph; (2) the simple semantic retriever underperforms compared to the prototype-based retriever. We attribute this difference to the fact that the prototype-based retriever can be fine-tuned based on **feedback from the downstream LM**, allowing for more effective optimization.

---

> ### Author Response · Authors · 2024-11-21
> **Response to Reviewer dzLA (2/3)**
>
> > **Q4.** Experiments are thin and lack detailed additional experiments and analysis, such as hyperparameters. Whether it can provide the performance of the model related to key hyperparameters, such as the number of nodes retrieved or different GNN models, is critical to understanding the key components of the method and the scalability.
>
> **A4.** We appreciate this great suggestion and are pleased to provide additional parameter studies. Following the reviewer's suggestion, we conducted the following experiments.
>
> 1. Firstly, we compared the performance of AugGLM equipped with GraphSAGE (used in the reported results) with the counterpart equipped with GCN [2]. The comparison is as follows:
>
> |  | Cora | Pubmed | ogbn-arxiv | ogbn-products |
> |---|---|---|---|---|
> | GraphSAGE | 91.14 | 94.80 | 75.39 | 81.73 |
> | GCN | 90.98 | 94.85 | 75.21 | 81.82 |
>
> We observed that the performance is **nearly identical** between GCN and GraphSAGE. This can be attributed to two factors: (1) the classification performances of GCN and GraphSAGE are **similar**, and (2) the GNN is used to generate prototypes and prune candidate labels, which **does not require a highly powerful GNN** for accurate classification.
>
>
> 2. Secondly, we preliminarily examined the relationship between the model performance and the number of nodes retrieved. In this auxiliary experiment, we fixed the number of nodes retrieved by semantic retrieval at $5$ and varied the number of nodes retrieved by PPR retrieval.
>
> | PPR nodes | 1 | 3 | 5 | 7 | 9 | 10 | 15 | 20 | 25 |
> |---|---|---|---|---|---|---|---|---|---|
> | ogbn-arxiv | 75.18 | 75.76 | 75.39 | 75.19 | 76.05 | 76.45 | 75.99 | 74.81 | 74.48 |
>
> Interestingly, we found that **the model's performance remains relatively stable when the number of PPR nodes is less than $15$**. However, the performance degrades when too many nodes are retrieved (more than $15$). A possible explanation is that **when the number of PPR nodes becomes too large, every target node's retrieved nodes become similar** (e.g., some hub nodes are retrieved by most nodes), reducing the discriminativeness of each target node. This phenomenon is reminiscent of the **"oversmoothing" problem** [3] in GNNs, where a GNN with too many layers produces indistinguishable latent representations for all nodes.
>
> We thank the reviewer for raising this insightful question. The above experiments are included in the revised Section E.2 (Appendix).
>
> ---
>
> > **Q5.** Most LM-based methods are inductive. Is the proposed method inductive? Have you tried zero-shot experiments? For example, training on one dataset. and test on other datasets.
>
> **A5.** We would like to address your question from the following perspectives:
>
> 1. Yes, our proposed framework is inductive **as long as the GNN used in our framework is also inductive** (e.g., GraphSAGE, which can be trained on the arxiv dataset before 2023 and then can be tested on the arxiv dataset after 2023).
>
> 2. Regarding whether our model can be trained on dataset $A$ and generalize to dataset $B$ without training on the dataset $B$, we answer that it **depends on how different the two datasets are**. The generalization of the LM is straightforward, as long as we formulate tasks on different datasets in **a shared text-to-text format**. However, our framework includes a GNN that is pre-trained on the given dataset, which might not be able to generalize across different datasets. We consider two cases: (1) **If the label space of
>  the two datasets is the same**, (e.g., arxiv before 2023 and arxiv after 2023), then the pre-trained GNN can generalize across these two datasets, and our framework can also generalize. (2) **If the label spaces are different** (e.g., arxiv and products, which have different numbers of labels and meanings for each one-hot label), then the GNN cannot generalize across different datasets, and our framework cannot generalize either.
>
> 3.  To our best knowledge, most node classifiers, such as [1], cannot perform zero-shot transfer learning. This is an interesting direction for future research in text-output node classifiers.

---

> ### Author Response · Authors · 2024-11-21
> **Response to Reviewer dzLA (3/3)**
>
> > **Q6.** Why wasn't the experiment conducted on decoder-only LLMs such as 7b model, which would have enhanced the integrity and significance of the paper. Although it doesn't change my rating on the paper.
>
> **A6.** We appreciate the reviewer's suggestion and acknowledge their reasonable evaluation that this does not detract from the quality of our paper. Our response to this question is as follows.
>
> 1. From a technical standpoint, our framework is **not limited to any specific downstream LMs** as long as they operate in a text-to-text manner. Therefore, we do not expect difficulties in generalizing our framework to decoder-only models such as Llama.
>
> 2. We included FLAN-T5s in our study because they represent **a diverse family of models with varying scales**. Their training is straightforward, and we believe our scientific goal has been achieved: showing the flexibility and superiority of the proposed AugGLM compared to the text-output SOTA models.
>
> 3. We did not include Llama in our study due to the **resource- and time-intensive** nature of its tuning, even when using parameter-efficient methods like LoRA.  For instance, the model's massive size necessitates small batch sizes for fine-tuning, which our lab **cannot accommodate due to limited GPU hours**. Given that our scientific goal has been met (as mentioned in point 2), we have opted not to include the Llama-version AugGLM in this paper, instead leaving it for future work.
>
> ```
> Reference:
>
> [1] Ye, Ruosong, et al. "Language is all a graph needs." EACL 2024.
>
> [2] Kipf, Thomas N., and Max Welling. "Semi-Supervised Classification with Graph Convolutional Networks." ICLR 2017.
>
> [3] Li, Qimai, Zhichao Han, and Xiao-Ming Wu. "Deeper insights into graph convolutional networks for semi-supervised learning." AAAI 2018.
>
> ```

---

> > ### Comment · Reviewer_dzLA · 2024-11-24
> > **Thanks for Your Response**
> >
> > Thanks for the detailed reply. It has addressed most of my concerns. But I also have some questions.
> > * I don't think "Language is all a graph needs." is the SOTA related work at the moment, for example, [1] has better performance on some datasets, and it is inductive and can work on multiple tasks.
> > * Regarding Q3, I observe that the performance of `Simple semantic retrieval` is strong, using an embedding model such as SimTeG [2] would have yielded better results, so I doubt that this paper's method would have any advantages.
> > * (Minor) In view of the responses in Q2 and Q5, I think the method in this paper is limited by the GCN model, which is less flexible than other pure LM-based approaches.
> >
> > [1] LLaGA: Large Language and Graph Assistant. In ICML 2024.
> >
> > [2] SimTeG: A Frustratingly Simple Approach Improves Textual Graph Learning.  In arxiv 2023.

---

> > > ### Author Response · Authors · 2024-11-25
> > > **Followup response (1/2)**
> > >
> > > Dear reviewer dzLA,
> > >
> > > We thank you for your continued engagement in the discussion phase. Our point-to-point responses to your questions are as follows.
> > >
> > > ---
> > >
> > > > **Additional Q1.** [1] has better performance on some datasets, and it is inductive and can work on multiple tasks.
> > >
> > >
> > > **Additional A1.** Thanks for suggesting LLaGA [1]. We will include it in our experimental baseline and related works. To clarify the differences between AugGLM and LLaGA, we would like to highlight the following perspectives:
> > >
> > > 1. Following [6], we use **a subset of the original ogbn-Products dataset** (Line 416), which differs from the ogbn-Products dataset used by **LLaGA**. Therefore, we compare the reported performance of the best variant of our AugGLM (with T5-base/large backbones) and the best variant of LLaGA (as per its Table 1) on the datasets *Cora, Pubmed, and ogbn-arxiv* as follows:
> > >
> > > |  | Cora | Pubmed | ogbn-arxiv |
> > > |---|---|---|---|
> > > | AugGLM | 91.51 | 95.16 | 76.80 |
> > > | LLaGA | 89.85 | 95.06 | 76.66 |
> > >
> > > It shows that our AugGLM **performs better** than LLaGA on these shared datasets for the node classification task.
> > >
> > > 2. We would like to emphasize that our work focuses on the **Text-to-Text** approach, which means we **do not make any changes to the overall architecture of the LM**. This makes our approach more versatile, as stated in Lines 40-49. In contrast, similar to soft prompting, LLaGA [1] **encodes the node embeddings into the output of the tokenization layer**, altering the original LM architecture.
> > >
> > > 3. Regarding the discussion of flexibility and generalization: We acknowledge that, as we responded to your initial Q2 and Q5, our method has limitations in zero-shot transfer learning when the label spaces of two datasets differ. This is because we encode graph information in the **input text space to preserve the Text-to-Text architecture** of the LM, and we believe that **a useful encoding strategy in the text space is to prune the candidate labels, which does not work in the zero-shot setting**. The flexibility of our approach lies in the fact that when **experimenting with or deploying different LMs**, we only need to set up the text input, text output, and instruct-tune the LM, **without requiring altering the LM's internal architecture**; In contrast, LLaGA's solution entails additional workload to modify the LM's architecture or source code accordingly.
> > >
> > > 4. We also appreciate the flexibility of LLaGA: once an LM is set up, it can potentially perform zero-shot transfer learning on multiple datasets/tasks. A possible modification to our framework to achieve this kind of flexibility would be to **inject the node embeddings from the GNN into an intermediate layer of the LM (e.g., the output of the tokenization layer)**, while retaining the topological and semantic retrieval modules. However, this would come at the cost of breaking the Text-to-Text architecture. Overall, we believe that our method and LLaGA explore two distinct and complementary paths.

---

> ### Author Response · Authors · 2024-11-25
> **Followup response (2/2)**
>
> > **Additional Q2.**
> Regarding Q3, I observe that the performance of Simple semantic retrieval is strong, using an embedding model such as SimTeG [2] would have yielded better results, so I doubt that this paper's method would have any advantages.
>
> **Additional A2.** We thank you for this suggestion. To show the effectiveness of our prototype-based semantic retriever, we conducted an additional ablation experiment as follows:
>
> 1. We replaced our prototype-based semantic retrieval module with a simple retriever (based on **node embeddings generated by SimTeG [2]**) that selects the $5$ most textually similar nodes. The remaining modules, including topological retrieval and classifier guidance, were left intact, and FLAN-T5-small is used as the LM backbone. In practice, we loaded the node embeddings *from the Huggingface repo [3] of SimTeg [2]* and performed inner product-based retrieval. The results are reported below.
>
> |  | ogbn-arxiv |
> |---|---|
> | Simple semantic retrieval | 74.46 |
> | SimTeG-based semantic   retrieval | 74.78 |
> | AugGLM | 75.39 |
>
> We found that the SimTeG-tuned retriever performs better on the node classification task but is still inferior to our prototype-based fine-tuned retriever. The reasons for this are as follows:
>
> 1. The training objective of the SimTeG-tuned retriever is to **align the classification loss with a GNN model [2]**, similar to knowledge distillation [5]. In other words, the SimTeG-tuned retriever is **a mixture of topological and semantic retrieval**, as the GNN incorporates both topology and node features. This means that its role **partially overlaps with that of the topological PPR retriever**.
>
> 2. Our prototype-based retriever can be **fine-tuned based on feedback from the downstream LM**, whereas the retrievers in the other two baselines cannot be fine-tuned in this way.
>
> Note that the backbone sentence encoder used for retrieval is the all-MiniLM-L6-v2. For the **Simple semantic retrieval** baseline, the retriever is the original one from Huggingface [4]; for the **SimTeG-based semantic retrieval** baseline, the retriever is tuned by SimTeG [3]; and for our model, the retriever is fine-tuned based on feedback from the downstream LM (FLAN-T5 in our paper).
>
> ---
>
> > **Additional Q3.**
> (Minor) In view of the responses in Q2 and Q5, I think the method in this paper is limited by the GCN model, which is less flexible than other pure LM-based approaches.
>
> **Additional A3.** We answered this in **Additional A1.**
>
> ---
>
> ```
> Reference:
>
> [1] LLaGA: Large Language and Graph Assistant. In ICML 2024.
>
> [2] SimTeG: A Frustratingly Simple Approach Improves Textual Graph Learning. In arxiv 2023.
>
> [3] https://huggingface.co/datasets/vermouthdky/SimTeG/tree/main/ogbn-arxiv/all-MiniLM-L6-v2/main/cached_embs
>
> [4] https://huggingface.co/sentence-transformers/all-MiniLM-L6-v2
>
> [5] Hinton, Geoffrey. "Distilling the Knowledge in a Neural Network." arXiv preprint arXiv:1503.02531 (2015).
>
> [6] He, Xiaoxin, et al. "Harnessing Explanations: LLM-to-LM Interpreter for Enhanced Text-Attributed Graph Representation Learning." The Twelfth International Conference on Learning Representations.
> ```
>
> As always, we would love to address any further questions you may have.
>
> Sincerely,
>
> Authors

---

> ### Author Response · Authors · 2024-12-02
>
> Dear reviewer dzLA,
>
> We appreciate your participation in the discussion phase. As the phase ends very soon, may we ask if our response to your follow-up questions is helpful?
>
> We are always happy to answer any questions and improve our manuscript.
>
> Sincerely,
>
> Authors

---

> > ### Comment · Reviewer_dzLA · 2024-12-02
> >
> > Thanks for your response and sorry for the late reply. Considering your feedback, and the discussion with other reviewers, I will keep my score and leave it for AC to decide,  the heavy coupling to GNN results in inflexibility and limited performance gains still exist.

---

> > > ### Author Response · Authors · 2024-12-03
> > >
> > > Dear reviewer dzLA,
> > >
> > > We appreciate your response and evaluation. Your main concern is that our framework is **heavily coupled to GNN, making it inflexible**. We want to recap the following facts regarding the flexibility comparison between our AugGLM and LLaGA [1].
> > >
> > > 1. Overall, our method and LLaGA explore two distinct solutions. LLaGA's text-based graph encoder and LM are cascaded in the embedding space, more like a conventional **model cascading**. Our framework uses a GNN to augment the text input of LM, more like a **text data augmentation** solution. These two solutions trade off flexibility in the following sense:
> > >
> > > 2. LlaGA **can achieve $0$-shot transfer learning**, e.g., training on the Cora dataset and testing on the Amazon dataset. However, LLaGA **cannot switch the LM seamlessly** because it requires modifying the LM's code and architecture, e.g., including its graph encoder's output into the LM's intermediate forward function.
> > >
> > > 3. Our AugGLM **cannot achieve $0$-shot transfer learning** because we need the GNN to provide reliable label candidates to augment LM input. However, thanks to such a data augmentation paradigm, AugGLM **can switch the LM seamlessly** as long as the LM works in a text-to-text manner. In addition, technically, our data augmentation-based solution **can work with black-box LM's fine-tuning API [2], but LLaGA cannot achieve that.**
> > >
> > > Regarding the performance gains, in our "Followup response (1/2)" and Table 1 in the paper, AugGLM is the **best text-output classifier on all the datasets**; moreover, AugGLM is **the best classifier**, even compared to the vector-output ones, on the **ogbn-products** datasets.
> > >
> > > We appreciate and respect your evaluation and are happy to invite AC and other reviewers to discuss and evaluate our method's flexibility or other limitations.
> > >
> > > Sincerely,
> > >
> > > Authors
> > >
> > > ```
> > > Reference:
> > >
> > > [1] Chen, Runjin, et al. "LLaGA: Large Language and Graph Assistant." Forty-first International Conference on Machine Learning.
> > >
> > > [2] https://platform.openai.com/docs/guides/fine-tuning/
> > > ```

---

### Official Review · Reviewer_GPZb · 2024-11-03

**Soundness:** 2
**Presentation:** 2
**Contribution:** 3
**Rating:** 6
**Confidence:** 3

**Summary:**

The paper presents a novel approach named AUGGLM that leverages off-the-shelf Language Models for node classification tasks on Text-Attributed Graphs (TAGs) without requiring any architectural modifications. The key contributions include: (1) Relevant Node Retrieval which utilizes topological and semantic retrieval methods to enrich the input of LMs with contextual information from relevant nodes. (2) Candidate Label Pruning which employs a lightweight GNN to prune class candidates, guiding the LMs' classification process. The approach maintains the original architecture of LMs, preserving their flexibility and efficiency in multi-task learning and personalized fine-tuning. The experiments on real-world datasets demonstrate that AUGGLM-equipped LMs outperform state-of-the-art text-output node classifiers and are comparable to top-performing vector-output node classifiers.

**Strengths:**

The approach creatively repurposes LMs for graph learning tasks, bridging the gap between specialized node classifiers and general LMs. This makes it more convenient for real-world applications as it does not require modifications to the LLM architecture. By maintaining the original architecture of LMs, the method preserves their versatility and compatibility with multi-task learning and personalized fine-tuning services.
The proposed topological and semantic retrieval methods effectively enhance the contextual information available to LMs, improving their performance on node classification tasks.
The paper provides thorough experimental validation on multiple datasets, showing consistent performance improvements.

**Weaknesses:**

Combining Figures 2 and 3 into a single figure would make it easier to understand the differences between the proposed approach and the InstructGLM method. This would provide a clearer picture of the main distinctions. Additionally, including an overall pipeline figure would help bridge the relationships among the various subsections mentioned in the methodology, offering a more cohesive understanding of the entire process.
The methods section is dense with rich content, making it challenging to follow. After reading the entire methods section, it is not easy to draw a complete picture of how the authors achieve their goal. A more streamlined presentation or a summary of key steps could improve clarity and comprehension.
The effectiveness of the retrieval-based augmentation heavily relies on the quality of the retrieved nodes. If the retrieval process fails to identify truly relevant nodes, the performance could degrade.
Due to the extensive details in the methods section, the authors have allocated limited space to present the experimental results and analyses. More discussions and deeper analyses of the experimental results would provide better insights into the performance and limitations of the proposed approach.

**Questions:**

How does the retrieval quality (both topological and semantic) impact the overall performance? Are there any metrics or evaluations to ensure the relevance of the retrieved nodes?
Can the authors provide a detailed analysis of the computational cost associated with the retrieval and candidate pruning steps? How does this compare to traditional GNN-based methods?
Have the authors tested the method on other types of graph datasets beyond TAGs?

---

> ### Author Response · Authors · 2024-11-21
> **Response to Reviewer GPZb**
>
> We appreciate the tremendous time and effort you took to review our paper. Thank you for recognizing that our proposed method is creative and effective, and that our experiments are thorough. Below, we provide point-by-point responses to your questions. Additionally, we have updated certain parts of the paper according to your suggestions, which are highlighted in **orange** font.
>
> ---
>
> > **Q1.** Combining Figures 2 and 3 into a single figure would make it easier to understand the differences between the proposed approach and the InstructGLM method. This would provide a clearer picture of the main distinctions.
>
> **A1.** Thanks for this great suggestion. We have incorporated the templates used by existing works and our work into a **revised Figure 3** in the pdf. We found that this suggestion indeed improved the readability of our paper.
>
> ---
>
> > **Q2.** Including an overall pipeline figure would help bridge the relationships among the various subsections mentioned in the methodology, offering a more cohesive understanding of the entire process. The methods section is dense with rich content, making it challenging to follow. After reading the entire methods section, it is not easy to draw a complete picture of how the authors achieve their goal. A more streamlined presentation or a summary of key steps could improve clarity and comprehension.
>
>
> **A2.** We appreciate this great suggestion. We have carefully illustrated a **detailed pipeline** in **revised Figure 2**, which provides a comprehensive overview of our proposed AugGLM.
>
> ---
>
> > **Q3.** The effectiveness of the retrieval-based augmentation heavily relies on the quality of the retrieved nodes. If the retrieval process fails to identify truly relevant nodes, the performance could degrade. How does the retrieval quality (both topological and semantic) impact the overall performance? Are there any metrics or evaluations to ensure the relevance of the retrieved nodes?
>
> **A3.** We agree with the reviewer that the quality of the retrieved nodes is crucial to the proposed framework, as mentioned in lines 213-214. We address your question regarding the two modules as follows.
>
> 1. For topological retrieval, we do not have a guarantee that retrieved nodes are relevant. However, our empirical results show that **PPR-retrieved nodes are of better quality than direct 1-hop neighbors**.
>
> |  | Cora | Pubmed | ogbn-arxiv | ogbn-products |
> |---|---|---|---|---|
> | 1-hop neigh. | 90.59 | 94.33 | 73.97 | 79.53 |
> | PPR | 91.14 | 94.80 | 75.39 | 81.73 |
>
> This is because PPR can explore the given graph **globally**, allowing it to capture more meaningful relationships between nodes. **The revised Section E. 2—"Other topological retrieval options"** provides a detailed study regarding the topological retriever.
>
> 2. For semantic retrieval, the prototype-based retriever is optimized using a **metric directly tied to the performance of the downstream LM** (Eqs. 12-14). This ensures that the retriever is trained to retrieve nodes most likely to improve the LM's ability to generate the groundtruth. To show the effectiveness of this optimized semantic retriever, we added a baseline: **a simple pre-trained semantic retriever (without fine-tuning with the LM)** that retrieves the $5$ most textually similar nodes as relevant nodes, while keeping the other modules unchanged (PPR retrieval and classifier guidance). The results are as follows:
>
> |  | Cora | Pubmed | ogbn-arxiv | ogbn-products |
> |---|---|---|---|---|
> | Simple semantic retrieval | 90.68 | 94.37 | 74.46 | 81.21 |
> | Prototype-based semantic retrieval | 91.14 | 94.80 | 75.39 | 81.73 |
>
> These results show that optimizing the retriever based on feedback from the LM can indeed ensure that the retrieved nodes are relevant.
>
> ---
>
> > **Q4.** Due to the extensive details in the methods section, the authors have allocated limited space to present the experimental results and analyses. More discussions and deeper analyses of the experimental results would provide better insights into the performance and limitations of the proposed approach.
>
> **A4.** We apologize that our analysis of the experiments is not satisfactory. We appreciate all the questions you raised in this review, and **most of them are included in the revised version** to provide a more comprehensive experimental section.
>
> ---
>
> > **Q5.** Have the authors tested the method on other types of graph datasets beyond TAGs?
>
> **A5.** No, we have not for the following reasons:
> 1. In this paper, we focus on the node classification tasks. The task is less interesting on other types of graphs, such as plain graphs.
>
> 2. It is well established that LMs **excel at understanding text**, and as we mentioned in Section 3, nodes in TAGs are associated with rich textual information. Therefore, in our opinion, TAGs provide the most suitable setting for investigating the expressiveness differences between GNNs and LMs, given that both models can effectively leverage node features.

---

> > ### Comment · Reviewer_GPZb · 2024-12-03
> > **Thanks for the clarification**
> >
> > Thank you for the efforts you put into the rebuttal. I have carefully reviewed your responses and found that my concerns have been addressed. While I have decided to maintain my original score, I think that the current contribution of the paper is marginally above the threshold. However, I am unable to assign a higher score at this time. But I remain supportive of the paper’s acceptance and look forward to its potential inclusion during the reviewer discussion phase.

---

> > > ### Author Response · Authors · 2024-12-03
> > > **Thank you**
> > >
> > > We are happy to know your concerns have been addressed, and thank you for your support and recognition!

---

### Official Review · Reviewer_iZTx · 2024-11-04

**Soundness:** 2
**Presentation:** 3
**Contribution:** 3
**Rating:** 6
**Confidence:** 4

**Summary:**

This paper introduces a novel framework for leveraging language models (LMs) in graph learning tasks. The key contributions include: (1) a dual-similarity node retrieval mechanism that integrates both topological and semantic features to identify relevant nodes, and (2) an innovative label space pruning technique that incorporates inductive biases derived from pre-trained graph neural networks (GNNs). The proposed approach demonstrates competitive performance across four diverse node classification benchmarks, suggesting its effectiveness as a general-purpose solution for graph-based learning tasks.

**Strengths:**

1. The paper presents an innovative approach to encoding graph structural information into a text format that is directly consumable by language models; This bridge between graph and textual domains enables leveraging the powerful capabilities of pre-trained LMs for graph-based tasks without requiring architectural modifications.



2. The proposed dual-similarity node retrieval system effectively captures both structural and semantic relationships; By considering both topology and semantics, the method provides richer contextual information to the LM compared to purely structure-based or content-based approaches


3. The integration of GNN-derived inductive biases through label space pruning represents a theoretically well-grounded approach; This technique effectively reduces the complexity of the classification task while preserving model performance.

**Weaknesses:**

1. The choice of Personalized PageRank (PPR) for topological similarity computation needs stronger justification. Modern GNNs with link prediction objectives could potentially provide more sophisticated structural representations. A comparative analysis between PPR and alternative approaches (e.g., GNN-based methods) would strengthen the methodological choices.


2. The title ``Language Models are Graph Learners'' suggests a broader scope than the actual focus on node classificatio. The method's generalizability claims would be more convincing with evaluation on: a). Link prediction tasks; b). Graph question-answering (GraphQA) scenarios. A discussion of potential adaptations needed for other graph learning tasks would be valuable.

3. The experimental evaluation lacks comprehensive comparisons with recent LLM-based graph learning methods[1-5]. A thorough comparison with these baselines would better contextualize the method's contributions and advantages.



[1]: Exploring the potential of large language models (llms) in learning on graphs
[2]: Graph chain-of-thought: Augmenting large language models by reasoning on graphs
[3]: GraphGPT: Graph Instruction Tuning for Large Language Models
[4]: Let Your Graph Do the Talking: Encoding Structured Data for LLMs
[5]: Graph Instruction Tuning for Large Language Models

**Questions:**

1. The performance metrics for GraphSAGE are absent from Table 2. There is a discrepancy between the methodology description and experimental results regarding GraphSAGE (line 420)

2. The paper claims to "retain the versatility of the original LM" (line 160) while requiring LM parameter tuning (line 372). How does the training process affect the LM's general capabilities? Is there evidence of catastrophic forgetting on the original LM tasks? Can the model generalize across different node classification domains (e.g., Arxiv to Products) without multi-task learning?


3. The experimental section would benefit from detailed efficiency metrics beside section 4.5. The metrics include FLOPs comparison with baseline methods, particularly InstructGLM, Memory usage analysis across different dataset scales, Convergence time comparisons, End-to-end runtime analysis including retrieval and inference phases.

4. Have the authors considered using GNNs with link-prediction objectives as an alternative to PPR?


5. For prototypical semantic retrieval, how is the performance of only relying on the semantic information (e.g. sentence representation based on SimCSE)?

---

> ### Author Response · Authors · 2024-11-21
> **Response to Reviewer iZTx (1/5)**
>
> Dear Reviewer iZTx, We appreciate the tremendous time and effort you took to review our paper. Thank you for recognizing our proposed method as innovative and reasonable. Below, we provide point-by-point responses to your questions.
>
> ---
>
> > **Q1.** The choice of Personalized PageRank (PPR) for topological similarity computation needs stronger justification. Modern GNNs with link prediction objectives could potentially provide more sophisticated structural representations. A comparative analysis between PPR and alternative approaches (e.g., GNN-based methods) would strengthen the methodological choices. Have the authors considered using GNNs with link-prediction objectives as an alternative to PPR?
>
> **A1.** Using a GNN-based link predictor to replace the role of PPR in our system is an **interesting yet challenging** approach. We would like to discuss the following perspectives with the reviewer.
>
> 1. Applying the **representation vectors** obtained from the link predictor to improve the LM is non-trivial. This is because (1) the LM can only process **text inputs**, not dense representation vectors, and (2) incorporating these vectors into the **intermediate layers of the LM** would require modifying its off-the-shelf architecture.
>
> 2. Using the link predictor to retrieve relevant neighbors is an interesting idea, and we have conducted additional experiments to verify this concept preliminarily. Specifically, we trained a **graph autoencoder (GA)** [7], a basic graph neural network-based link predictor, on the given graph. Then, we retrieved the top-$5$ most confident neighbors from the **reconstructed graph** to replace those obtained through **PPR retrieval**. The results are presented in the table below, where Flan-T5-small is used as the backbone. For better reference, we also provide a version where PPR retrieval is replaced with retrieving from $1$-hop neighbors.
>
> |  | Cora | Pubmed | ogbn-arxiv | ogbn-products |
> |---|---|---|---|---|
> | 1-hop neigh. | 90.59 | 94.33 | 73.97 | 79.53 |
> | GA | 90.83 | 94.42 | 74.01 | 79.85 |
> | PPR | 91.14 | 94.80 | 75.39 | 81.73 |
>
> We observe that both $1$-hop neighbor retrieval and GA perform worse than their PPR counterparts. A possible reason is that both $1$-hop neighbor retrieval and GA are **local** retrieval methods, whereas PPR can capture the **global** structure effectively. Additionally, we note that GA is trained using a reconstruction loss, which means it **tends to assign high confidence to existing edges**. In other words, the neighbors retrieved by GA would be similar to those obtained through $1$-hop neighbor retrieval, except for some low-degree nodes.
>
> 3. Lastly, we strongly agree with the reviewer that using link predictors for topological retrieval is a promising idea, and the key challenge lies in finding the optimal balance **between local and global retrieval**. We leave a systematic study of this topic as future work.
>
> ---
>
> > **Q2.** The title ``Language Models are Graph Learners'' suggests a broader scope than the actual focus on node classification. The method's generalizability claims would be more convincing with evaluation on: a). Link prediction tasks; b). Graph question-answering (GraphQA) scenarios. A discussion of potential adaptations needed for other graph learning tasks would be valuable.
>
>
> **A2.** We appreciate this suggestion. For the current work, we are happy to change the title of this paper to "Language Models are Node Classifiers." Additionally, we would like to discuss potential adaptations as follows:
>
> 1. For **link prediction** tasks, the adaptation can be straightforward. For example, (1) we can use our proposed topological retrieval and semantic retrieval to retrieve neighbors for each node from the **target node pair**, and (2) we can utilize a **pretrained link predictor** to guide the prediction of the LM, by using a template such as *"an expert model predicts that this node pair is connected with percentage xx \%"*.
>
> 2. For **GraphQA**, if the reviewer refers to knowledge graph QA [6], adapting our method to this task requires **additional effort**. For example, the semantic retrieval might need to **incorporate edge texts** to better understand the **triplets** from the knowledge graph. We believe that such an adaptation is highly interesting and may warrant an independent study.
>
> The above discussion will be included in the "Future Work" section of the revised version.

---

> ### Author Response · Authors · 2024-11-21
> **Response to Reviewer iZTx (2/5)**
>
> > **Q3.** The experimental evaluation lacks comprehensive comparisons with recent LLM-based graph learning methods[1-5]. A thorough comparison with these baselines would better contextualize the method's contributions and advantages. [1]: Exploring the potential of large language models (llms) in learning on graphs [2]: Graph chain-of-thought: Augmenting large language models by reasoning on graphs [3]: GraphGPT: Graph Instruction Tuning for Large Language Models [4]: Let Your Graph Do the Talking: Encoding Structured Data for LLMs [5]: Graph Instruction Tuning for Large Language Models
>
> **A3.** Thank you for referring us to these valuable references. We were aware of the above papers and would like to provide some context to clarify their relevance to our work. Overall, (1) we selected baselines from **leaderboards (lines 422 and 423)**, and (2) we would like to highlight the **unique perspectives from each of them** so that the reviewer can better understand our paper's position. Furthermore, we are happy to include them in the related work section for a more comprehensive literature review. Here is a brief summary of each paper:
>
> 1. Our paper: We focus on the problem of **node classification on text-attributed graphs (TAGs)**, where the LM is **fine-tuned**, and its architecture is completely **preserved**.
>
> 2. [1]: This paper also addresses **node classification on TAGs** but it is more of a systematic exploration that includes many existing works such as TAPE and KEA. Moreover, the LM used in this paper is **not fine-tuned**. Its reported performance is not superior to our model, and we included it in the revised related work section.
>
> 3. [2]: This paper studies **graph reasoning** which is a **different problem from ours**. Therefore, we did not select it as a baseline. However, we included it in the revised related work section.
>
> 4. [3]: This paper tackles both **node classification and link prediction on TAGs**. However, it heavily modifies the architecture of the LM and focuses more on the **transfer learning** setting. The only results that could be included in our paper are the column "arxiv-arxiv" in Table 1, whose performance (ACC, 75.11) is worse than our reported 76.80. This work was included in the related work section of the original submission.
>
> 5. [4]: This paper studies **graph reasoning on plain graphs** (e.g., node counting) which is a **different problem from ours**. Therefore, we did not select it as a baseline. However, we included it in the revised related work section.
>
> 6. [5]: If we understood correctly, this is the same work as [3], and was included in the related work section of the original submission.
>
>
> ---
>
> > **Q4.** The performance metrics for GraphSAGE are absent from Table 2. There is a discrepancy between the methodology description and experimental results regarding GraphSAGE (line 420)
>
> **A4.** We believe there has been a misunderstanding. Line 420 states that **in our framework**, the GNNs used for **generating prototypes** (Eq. 6) and **classifier guidance** (Eq. 9) are GraphSAGE. In other words, GraphSAGE is used in all variants of our AugGLM model (T5-small/base/large).
>
> However, we appreciate the suggestion to include GraphSAGE as a baseline in Table 2. To address this, we report its performance on Cora/Pubmed/ogbn-arxiv/ogbn-products as follows: 86.51/89.08/73.88/76.04. They have been incorporated into the revised Table 1.

---

> ### Author Response · Authors · 2024-11-21
> **Response to Reviewer iZTx (3/5)**
>
> > **Q5.** The paper claims to "retain the versatility of the original LM" (line 160) while requiring LM parameter tuning (line 372). How does the training process affect the LM's general capabilities? Is there evidence of catastrophic forgetting on the original LM tasks? Can the model generalize across different node classification domains (e.g., Arxiv to Products) without multi-task learning?
>
> **A5.** We would like to address this question from the following perspectives:
>
> 1. We apologize for any confusion- when we mentioned **"versatility"**, we were referring to **the ability to be fine-tuned on various tasks/datasets**. This is empirically verified in our multi-task training (section 5.4).
>
> 2. Regarding whether the model forgets the original LM tasks, we have two points to make: (1) it is challenging to systematically verify this because the FLAN-T5 models used in this paper are pretrained on **a vast number of tasks [8]**, making verification very challenging. (2) To our knowledge, most LM finetuning solutions **do not prioritize retaining the model's performance on original LM pretraining tasks**. For example, [9] (LM fine-tuned for graphs) and [10] (LM fine-tuned for text ranking) do not evaluate their fine-tuned LMs' performance on the original LM pretraining task.
>
> 3. Regarding whether our model can be trained on dataset $A$ and generalize to dataset $B$ without training on dataset $B$, we answer that it **depends on how different datasets $A$ and $B$ are**. The generalization of the LM is straightforward as long as we formulate tasks on different datasets in **a shared text-to-text format**. However, our framework includes a GNN pretrained on the given dataset, which might not generalize across different datasets. We outline the two cases here: (1) **If the label spaces of the two datasets are the same**, e.g., arxiv before 2023 and arxiv after 2023, then the pretrained GNN can generalize across these two datasets, and our framework can also generalize. (2) **If their label spaces are different**, e.g., arxiv and products, whose number of labels and the meaning of every one-hot label are entirely different, then the GNN cannot generalize across different datasets, and hence, our framework cannot generalize either.

---

> ### Author Response · Authors · 2024-11-21
> **Response to Reviewer iZTx (4/5)**
>
> > **Q6.** The experimental section would benefit from detailed efficiency metrics beside section 4.5. The metrics include FLOPs comparison with baseline methods, particularly InstructGLM, Memory usage analysis across different dataset scales, Convergence time comparisons, End-to-end runtime analysis including retrieval and inference phases.
>
> **A6.** We appreciate  this great suggestion. Following up on this, we conducted additional experiments to evaluate the efficiency of our proposed framework.
>
> 1. *FLOPs*: The computation of *InstructGLM* includes (1) computing node encodings, which is **precomputed**, and (2) **training and inference** of the downstream **LM**. In contrast, the computation of our *AugGLM* includes (1) computing PPR neighbors for every node, which is also **precomputed**, (2) **training and inference** of the **semantic retriever**, and (3) **training and inference** of the downstream **LM**. Hence, the extra on-the-fly computation overhead comes from the semantic retriever, all-MiniLM-L6-v2, in our experiments. We report the FLOPs of the retriever and different LM backbones as follows,
>
> |  | FLOPs |
> |---|---|
> | Retriever | 2.3G |
> | FLAN-T5 (small) | 71.7G |
> | FLAN-T5 (base) | 257.2G |
> | FLAN-T5 (large) | 845.4G |
>
> The results show that the retriever only adds a tiny amount of FLOPs compared to the backbone LMs. In other words, **the FLOPs of our framework are very close to those of InstructGLM** if we adopt the same downstream LM. More concretely, **if both our model and InstructGLM select T5-large as the backbone, the FLOPs of InstructGLM would be $845.4$G, and our framework would be $847.7$G.**
>
> 2. *Memory usage*: Memory usage is linear concerning batch size. We report the memory usage with different backbone LMs as follows, where we set the batch size to $1$.
>
> |  | GPU memory |
> |---|---|
> | AugGLM (small) | 3098M |
> | AugGLM (base) | 6572M |
> | AugGLM (large) | 20308M |
>
> It is reasonable that more powerful backbone LMs require more GPU memory.
>
> 3. *Convergence analysis*: **In Section E.1-"Convergence analysis" of the revised version**, we updated the convergence curve of our proposed AugGLM with backbone LMs as FLAN-T5-small/base/large on the Cora dataset. The results show that our proposed model's training is **smooth** with backbone **LMs of different scales**.
>
> 4. *End-to-end running time*: we recorded the running time (both forward and backpropagation) of the semantic retriever and the backbone LMs in the following table. The dataset we tested was Cora, and the batch size was 1. Note that this wall clock running time is related to the batch size, dataset, and specific hardware. Overall, we can conclude that **the semantic retriever only adds very limited on-the-fly computation overhead compared to the downstream LM**, showing the efficiency of our proposed framework.
>
> |  | Forward (ms) | Backprop (ms) |
> |---|---|---|
> | Retriever | 14.7 | 6.1 |
> | FLAN-T5 (small) | 90.0 | 32.0 |
> | FLAN-T5 (base) | 104.4 | 66.6 |
> | FLAN-T5 (large) | 277.2 | 197.0 |
>
> **All the above results are included in the revised paper in Section E.1 (Appendix).**
>
> ---
>
> > **Q7.** For prototypical semantic retrieval, how is the performance of only relying on the semantic information (e.g. sentence representation based on SimCSE)?
>
> **A7.** We appreciate this excellent suggestion. To answer it, we implemented the following baseline: we directly used a pre-trained sentence encoder to retrieve the top-$5$ semantically similar nodes for every node. Note that such a simple semantic retriever is **intractable to fine-tune based on feedback from the downstream LM** because the number of possible combinations is enormous; allowing the LM to make inferences for every possible combination is computationally infeasible.
>
> |  | Cora | Pubmed | ogbn-arxiv | ogbn-products |
> |---|---|---|---|---|
> | Simple semantic retrieval | 90.68 | 94.37 | 74.46 | 81.21 |
> | Prototype-based retrieval | 91.14 | 94.80 | 75.39 | 81.73 |
>
> The results show that the simple semantic retriever's performance is inferior to that of the prototype-based retriever. We believe the difference is due to the prototype-based retriever's ability to fine-tune based on feedback from the downstream LM.

---

> ### Author Response · Authors · 2024-11-21
> **Response to Reviewer iZTx (5/5)**
>
> ```
> Reference:
>
> [1] Chen, Zhikai, et al. "Exploring the potential of large language models (llms) in learning on graphs." ACM SIGKDD Explorations Newsletter 25.2 (2024): 42-61.
>
> [2] Jin, Bowen, et al. "Graph chain-of-thought: Augmenting large language models by reasoning on graphs." arXiv preprint arXiv:2404.07103 (2024).
>
> [3] Tang, Jiabin, et al. "Graphgpt: Graph instruction tuning for large language models." SIGIR 2024.
>
> [4] Perozzi, Bryan, et al. "Let your graph do the talking: Encoding structured data for llms." arXiv preprint arXiv:2402.05862 (2024).
>
> [5] Tang, Jiabin, et al. "Graphgpt: Graph instruction tuning for large language models." SIGIR 2024.
>
> [6] Huang, Xiao, et al. "Knowledge graph embedding based question answering." WSDM 2019.
>
> [7] Kipf, Thomas N., and Max Welling. "Variational graph auto-encoders." arXiv 2016.
>
> [8] Chung, Hyung Won, et al. "Scaling instruction-finetuned language models." JMLR 2024.
>
> [9] Ye, Ruosong, et al. "Language is all a graph needs." EACL 2024.
>
> [10] Zhuang, Honglei, et al. "Rankt5: Fine-tuning t5 for text ranking with ranking losses." SIGIR 2023.
> ```

---

> ### Author Response · Authors · 2024-11-23
> **Additional Experiment: Adaptation to Link Prediction Task for Q2.**
>
> > **Q2.** The method's generalizability claims would be more convincing with evaluation on: a). Link prediction tasks; b). Graph question-answering (GraphQA) scenarios. A discussion of potential adaptations needed for other graph learning tasks would be valuable.
>
> We would like to thank you again for this great suggestion, which we believe will significantly improve the impact and quality of our paper.
>
> A discussion regarding the possible adaptations to link prediction and GraphQA has been included in our answer **A2**. Here, to address your question **Q2** further, we conducted an additional experiment to adapt our proposed AugGLM to the **link prediction** task. Link prediction can be viewed as **a classification task for a pair of nodes**. For all modules, we made the following adaptations:
>
> 1. We retained the topological PPR retrieval for the **input node pair**.
>
> 2. We **concatenated the text of the node pair** as input for the semantic retriever. The prototypes used as the corpus of the semantic retriever were still generated by a pre-trained GNN, which is consistent with our approach for the node classification task.
>
> 3. For classifier guidance,  we utilized a pre-trained graph autoencoder (GAE), whose output is the **connection probability** for every node pair. We **transformed the connection probability into plain language** based on the following rules: (1) less than 0.2: "improbable", (2) 0.2 to 0.4: "unlikely", (3) 0.4 to 0.6: "maybe", (4) 0.6 to 0.8: "likely", and (5) more than 0.8: "highly likely". The GAE's prediction (in plain language) was then incorporated into the following template.
>
> 4. The template we used is in the following format:
>
> ```
> Please determine if the following two papers are related or not.
> Paper 1's title: {Paper 1's title}\nPaper 1's abstract: {Paper 1's abstract}\nPaper 1's related works: {Paper 1's PPR neighbors' titles}.
>
> Paper 2's title: {Paper 2's title}\nPaper 2's abstract: {Paper 2's abstract}\nPaper 2's related works: {Paper 2's PPR neighbors' titles}.
>
> Other related works: {Semantic retrieved nodes' titles}.
>
> An expert link prediction model predicted that the possibility of these two papers being related is: {GAE's prediction}. Do you think these two papers are related or not? Please answer Yes or No.
> ```
>
> We conducted preliminary experiments on the Cora dataset, following the settings from the benchmark [1]. In this setup, 5\% and 10\% of edges were removed for validation and testing, respectively. Also, an equal number of non-connected node pairs were used as negative samples. The accuracy results are reported in the following table.
>
> |  | Accuracy |
> |---|---|
> | GAE | 89.29 |
> | AugGLM (small) | 93.59 |
> | AugGLM (base) | 94.25 |
>
>
> Our key findings are as follows:
>
> 1. Our proposed AugGLM can indeed be effectively adapted to link prediction tasks.
>
> 2. By leveraging a classic link predictor (GAE), our AugGLM achieves a significant performance boost over the backbone predictor, which aligns with our observations in node classification tasks.
>
> We will include more comprehensive experiments and additional baselines for the Link Prediction task in the camera-ready version.
>
> [1] https://paperswithcode.com/paper/variational-graph-auto-encoders

---

> ### Comment · Reviewer_iZTx · 2024-11-24
> **Response to authors**
>
> Having thoroughly addressed all the concerns raised in my initial review, the authors have significantly strengthened the paper. Based on these comprehensive improvements, I will increase the scores accordingly.

---

> ### Author Response · Authors · 2024-11-24
>
> Thank you. We are glad to hear that your concerns have been addressed and your suggestions help improve our work greatly. As always, we would love to address any further questions you may have.

---

### Author Response · Authors · 2024-11-21
**General Response**

Dear reviewers,

Thank you for taking the time to provide valuable reviews of our paper. We appreciate your recognition of our method as innovative and creative (iZTx and GPZb) and your acknowledgment that the proposed method is effective (iZTx and GPZb). Additionally, we are grateful for your positive comments on the organization of our paper (dzLA) and the comprehensiveness and reliability of our experiments (GPZb and dzLA). We also thank you for your constructive comments on how to improve this paper further. We made the following substantial improvements in the *revised paper*, all marked in **orange** font in the pdf file.

1. Presentation: (1) Two templates are merged into **Figure 3** for a side-by-side comparison. (2) A detailed pipeline is provided in **Figure 2** to illustrate our proposed modules.

2. Experiments: (1) Section E.1 (appendix) provides **additional efficiency studies**, including **FLOPs comparison**, **memory usage**, **convergence analysis**, and **wall-clock running time**. (2) Section E.2 provides **additional parameter study**, including the selection of **GNNs**, **number of retrieved nodes**, and the **other topological retriever** such as graph autoencoder.

3. Related works: More related works are included in Section 2 about **"LM for Graphs."**


Below, we provide detailed responses to your questions and concerns. We sincerely invite you to engage in further discussion with us.

Authors

---

### Meta-Review · Area_Chair_uD3h · 2024-12-20

**Metareview:**

## Summary of Scientific Claims and Findings
The paper introduces AUGGLM (Augmented Graph Language Model), a framework that leverages off-the-shelf language models (LMs) for node classification tasks on Text-Attributed Graphs (TAGs). Key components include dual-similarity node retrieval mechanisms using topological and semantic features, and a label space pruning technique employing a lightweight Graph Neural Network (GNN). The approach aims to maintain the original architecture of LMs, thus preserving their versatility and compatibility with multi-task learning. The proposed method demonstrates competitive performance across several node classification benchmarks.

## Strengths
1. **Innovative Approach**: The paper creatively applies language models to graph learning tasks, which is a relatively novel intersection of domains.
2. **Preservation of LM Architecture**: By maintaining the original LM architecture, the method ensures versatility and broader applicability.
3. **Comprehensive Experiments**: The paper includes a wide range of experiments across multiple datasets, supporting the claimed effectiveness of the method.

## Weaknesses
1. **Lack of Novelty**: Reviewers noted that the method primarily combines existing techniques without introducing significant novel contributions. The approach appears to be a combination of well-established methods rather than a groundbreaking new technique.
2. **Complexity and Clarity**: The methodology section is dense and complex, making it difficult for readers to follow the flow and understand the contribution. The rationale for integrating the different modules is not clearly articulated.
3. **Limited Generalizability**: The method's reliance on a GNN component could limit its flexibility and the potential for zero-shot or cross-domain applications.
4. **Insufficient Theoretical Justification**: The paper lacks a strong theoretical foundation for the integration of its components, leaving the impression that the system is a compilation of known techniques rather than a cohesive new method.

After thorough consideration of other submissions in the same batch, I find myself recommending rejection for this paper. I encourage the authors to refine their work and consider resubmission to a future conference or journal.

**Additional Comments On Reviewer Discussion:**

## Summary of Discussion and Rebuttal
- **Reviewers' Points**: Concerns were raised about the novelty of the contributions, the complexity of the presentation, and the limited generalizability due to the GNN dependency. Reviewers also noted the lack of clear theoretical justification for the proposed system.
- **Authors' Responses**: The authors provided additional explanations and conducted further experiments, addressing some of the reviewers' specific questions. However, the responses did not sufficiently resolve the primary concerns about novelty and clarity.

In conclusion, while the paper explores an interesting intersection of language models and graph learning, the lack of significant novelty, combined with generalizability concerns. After thorough consideration of other submissions in the same batch, I recommend rejection for this paper. Future submissions could benefit from a clearer focus on novel contributions and improved clarity in presenting the methodology.

---

### Decision · Program_Chairs · 2025-01-22

Reject